# Endorsement of TNBC Biomarkers in Precision Therapy by Nanotechnology

**DOI:** 10.3390/cancers15092661

**Published:** 2023-05-08

**Authors:** Aiswarya Chaudhuri, Dulla Naveen Kumar, Deepa Dehari, Rohit Patil, Sanjay Singh, Dinesh Kumar, Ashish Kumar Agrawal

**Affiliations:** 1Department of Pharmaceutical Engineering and Technology, Indian Institute of Technology (BHU), Varanasi 221005, India; 2Department of Pharmaceutics, Babasaheb Bhimrao Ambedkar University (A Central University), Vidya Vihar, Raebareli Road, Lucknow 226025, India

**Keywords:** triple-negative breast cancer, biomarkers, nanoparticles, targeted therapy, personalized therapy

## Abstract

**Simple Summary:**

Triple-negative breast cancer (TNBC), the most aggressive and heterogenous type of cancer, lacks the expression of hormones like estrogen, progesterone, and human epidermal growth receptor-2, making chemotherapy the only treatment regimen against TNBC. It was further observed that chemotherapy leads to off-site toxicity and chemoresistance, decreasing anticancer activity. Therefore, to overcome the problem faced during chemotherapy, and to address the heterogeneity of TNBC, targeted therapy emerged based on the molecular profiling of TNBC. Such a scenario further encouraged the acceptance of biomarkers as some of the targeting moieties for effective and precise TNBC therapy. Several biomarkers are used as targets for the precision therapy in TNBC, including EGFR, VGFR, TP53, interleukins, insulin-like growth factor binding proteins, c-MET, androgen receptor, BRCA1, glucocorticoid, PTEN, ALDH1, etc. Additionally, nanoparticles were employed as parts of a multifunctional platform to deliver the therapeutics to the target site with increased precision.

**Abstract:**

Breast cancer is a heterogeneous disease which accounts globally for approximately 1 million new cases annually, wherein more than 200,000 of these cases turn out to be cases of triple-negative breast cancer (TNBC). TNBC is an aggressive and rare breast cancer subtype that accounts for 10–15% of all breast cancer cases. Chemotherapy remains the only therapy regimen against TNBC. However, the emergence of innate or acquired chemoresistance has hindered the chemotherapy used to treat TNBC. The data obtained from molecular technologies have recognized TNBC with various gene profiling and mutation settings that have helped establish and develop targeted therapies. New therapeutic strategies based on the targeted delivery of therapeutics have relied on the application of biomarkers derived from the molecular profiling of TNBC patients. Several biomarkers have been found that are targets for the precision therapy in TNBC, such as EGFR, VGFR, TP53, interleukins, insulin-like growth factor binding proteins, c-MET, androgen receptor, BRCA1, glucocorticoid, PTEN, ALDH1, etc. This review discusses the various candidate biomarkers identified in the treatment of TNBC along with the evidence supporting their use. It was established that nanoparticles had been considered a multifunctional system for delivering therapeutics to target sites with increased precision. Here, we also discuss the role of biomarkers in nanotechnology translation in TNBC therapy and management.

## 1. Introduction

Triple-negative breast cancer (TNBC) is the most aggressive subtype of breast cancer, entailing no expression of estrogen receptor, progesterone receptor, and human epidermal receptor-2. It accounts for approximately 10–20% of total breast cancer cases and is found to be most prevalent in young African and Hispanic women [1]. According to the American Cancer Society and the National Cancer Institute, in 2020, approximately 276,480 new cases of TNBC occurred, wherein almost 42,170 women died [2]. TNBC is considered aggressive due to its heterogeneity, rapid metastasizing ability to the brain, lungs, and bones, and rapid onset of recurrence [3], which makes the treatment regimen difficult for TNBC. Moreover, as TNBC lacks the expression of hormones, endocrine therapy is out of the option, making chemotherapy the only treatment against TNBC [4]. From the molecular profiling, it came into focus that there are six molecular subtypes of TNBC, which include basal-like subtypes (BL1 and BL2), mesenchymal (M), mesenchymal stem-like (MSL), immunomodulatory (IM), and luminal androgen receptor (LAR) [5]. Further, on performing the genetic profiling of the molecular subtypes, it was found that these subtypes show either aberrant genetic expression or highly activated signaling pathways or receptors. For example, BL1 and BL2 subtypes show aberrant expressions of DNA-repair and cell-cycle regulating genes like MYC, PIK3CA, AKT2, CDK6, and BRCA2, and PTEN, RB1, and TP53, respectively. Similarly, MSL subtypes also show an aberrant expression of genes related to cell proliferation and stemness (ALDHA1, BCL2, BMP2, HOX, etc.) On the other hand, M and IM subtypes exhibit highly activated signaling pathways like Wnt, TGF-β, NK cell, IL-12, IL-7, etc. Moreover, LAR subtypes show highly activated androgen hormone-related signaling pathways [6]. It was thus inferred that the heterogeneous nature of TNBC might compromise the therapeutic efficacy of the chemotherapy.

Moreover, the conventional neoadjuvant chemotherapy exhibited pCR in 35–45% of TNBC patients only, and a majority of the TNBC patients showing responsiveness to conventional chemotherapy were limited to the non-metastatic stage [7]. Thus, to overcome such problems and make the treatment more precise, biomarkers have emerged as targeted therapeutic and diagnostic tools. Scientists are using cancer biomarkers to acquire knowledge regarding patients’ tumors to predict the personalized treatment regimens specific to particular TNBC subtypes. These predictive biomarkers include various germline and somatic mutations, genetic rearrangements, proteins, and metabolomics [8]. However, it was observed that none of the biomarkers achieve 100% in both sensitivity as well as specificity [9], and also that as the cancer treatment implements more combination therapy as compared to monotherapy, it becomes difficult to attach an identified biomarker with a single drug or target [10]. Hence, to increase specificity and to efficiently deliver multiple diagnostic and therapeutic molecules to a target site, nanoparticles (NPs) were developed based on their exclusive physiochemical characteristics. It was further observed that for improved sensitivity, and targetability, NPs are modulated to incorporate cancer-specific ligands having increased binding affinities towards TNBC biomarkers [9].

In this review, we discuss well-established TNBC biomarkers and explore nanoparticle-based technologies employed for increasing the sensitivity and specificity of the biomarkers at a targeted site. We also discuss the ongoing clinical trials on these biomarkers, and biomarkers-based nanoparticles, which are employed as therapeutic and diagnostic tools against TNBC.

## 2. Biomarkers Derived from the Molecular Profiling of TNBC

Biomarkers are classified as reproducibly quantifiable biological variables. As defined by the National Institute of Health, clinically, they are considered measurable parameters used to evaluate the responses offered by therapeutic interventions. Additionally, they can be regarded as factors employed for early diagnosis, monitoring, and personalized treatment [11].

As discussed above, although in some TNBC patients targeted therapy does receive certain clinical benefits, the overall responses in TNBC patients remain limited. Such a scenario urges the need to develop more robust targeted approaches for improving the therapeutic outcomes in TNBC patients. The various biomarkers investigated as potential targets for TNBC include intracellular signalings like kinases, cell cycle, cell death regulation, and DNA damage (Figure 1) [12].

In recent times, a series of TNBC biomarkers have been evaluated. These TNBC biomarkers can be classified based on their usages as prognostic (biomarkers giving information regarding the overall outcome, regardless of the therapy), predictive (biomarkers providing information on the effect of a therapeutic intervention), and diagnostic (biomarkers confirming the presence of the disease) [13], based on the site where the biomarkers are found to be available like in the blood, cytoplasm, and nucleus, and on the surface of cells [10], or based on target expression in DNA, RNA, and proteins [14]. One biomarker can simultaneously be prognostic, predictive, and diagnostic [15]. It becomes reasonable to classify biomarkers based on the site where they are found, as this will aid in developing a strategy suitable for delivering the appropriate diagnostic and therapeutic moiety to the target site more efficiently. Moreover, if their area of availability is known, we can modulate or personalize the delivery or targeting system by changing their nature and characteristics.

### 2.1. TNBC Biomarkers on the Cell Surface

#### 2.1.1. Folate Receptor

Cancer exhibits the overexpression of specific receptors, sometimes recognized as the biomarkers used to diagnose cancer. Folate receptor alpha (FRα) is one of the well-recognized prognostic biomarkers employed for the diagnosis of TNBC [16]. At the molecular level, folate plays an essential role in cell metabolism. FRα is a glycosylphosphatidylinositol (GPI) membrane-bound protein that exhibits increased binding affinity with folate, facilitating the increased transportation of folate into the cells. Hence, it was observed that the overexpression of FRα confers tumor growth via increased folate uptake, which may alter specific cellular signaling pathways and cause enhanced cell proliferation [17]. It was also found that folate sustains metabolic reactions, creating a suitable tumor microenvironment (TME) essential for the growth of cancer cells. Various genomic studies have revealed that 30% of the early-staged TNBC cases show the overexpression of FRα, whereas FRα is found to be overexpressed in 70–80% of stage IV metastatic TNBC cases [16].

#### 2.1.2. Epidermal Growth Factor Receptor (EGFR)

The epidermal growth factor receptor (EGFR) belongs to the ErbB family of receptor tyrosine kinases (RTKs), delivering pivotal functions in cell physiology. EGFR was found to be overexpressed or mutated in 13–76% of TNBC cases. EGFR is an example of a prognostic biomarker [14,18,19]. In unstimulated conditions, EGFR remains dimerization-incompetent and auto-inhibited at the plasma membrane. On binding with the ligand, the receptor gets activated allosterically, undergoes dimerization, and facilitates autophosphorylation of the tyrosine residue, which finally triggers signaling cascades like cell growth, proliferation, metastasis, and angiogenesis [19].

#### 2.1.3. Interleukin-3—Receptor α (IL-3Rα)

Interleukin-3 (IL-3) is a cytokine that is comprised of a heterodimeric receptor, consisting of an α-chain, which is the specific binding subunit, and a common β-chain, which is shared with GM-CSF (granulocyte-macrophage colony-stimulating factor) and IL-5 (interleukin-5) receptors. IL-3 is generated via activated T cells and mast cells and is associated with regulating hemopoietic pluripotent and expanding progenitor cells. It is a predictive biomarker. It was observed that on binding with its receptor, the IL-3 proceeds various biological processes such as the expression of adhesion molecules, proteins, and inflammatory and transcriptional factors. In addition, IL-3 plays a role in regulating cell survival and the proliferation of tumor-derived endothelial cells (TEC) and increasing the expression of the AKT signaling pathway and pro-tumorigenic and angiogenic receptors, thereby controlling the tumor microenvironment. From various clinical trials, it was observed that 55% of TNBC patients showed overexpression of IL-3R α. It was further observed that in TNBC, TECs release extracellular vesicles responsible for cell invasion, epithelial-to-mesenchymal transition (EMT), vascular mimicry (VM), and metastasis to secondary sites like the brain, bone, and lungs [20,21].

#### 2.1.4. c-Kit

c-Kit, also known as CD117, is the receptor tyrosine kinase (RTK) for the stem cell factor (SCF), which is encoded by the proto-oncogene *c-Kit* situated on the 4q12 chromosome [22,23]. It is a prognostic biomarker. It was observed that c-Kit signaling plays a pivotal role in cellular differentiation [24], and that its gain-of-function mutation leads to the activation of various downstream pathways like the PI3K/AKT/mTOR, MAPK, and JAK/STAT transduction pathways [25]. It was observed that 25–45% of cases TNBC exhibits overexpression of c-Kit [22,26,27]. Earlier, sunitinib was employed against c-Kit-induced TNBC. Still, no improved results were observed, leading to the development of a new tyrosine kinase inhibitor that can block the functioning of c-Kit. Currently, dasatinib, sorafenib, and Nilotinib are used commercially to treat TNBC that exhibits overexpression of c-Kit [22,26].

#### 2.1.5. c-Met

Similar to c-Kit, c-Met is also a prognostic biomarker. c-Met, also known as hepatocyte growth factor receptor (HGFR), is also a receptor tyrosine kinase that is encoded by *c-Met* proto-oncogene. It was observed that the activation of c-Met proto-oncogene via mutation, amplification, enhanced transcription, and increased ligand activation leads to increased cell growth, proliferation, invasion, and migration [28]. In ligand-activation mode, c-Met gets activated by binding with HGF in a paracrine method [29]. Such binding results in the autophosphorylation of tyrosine residues present within the kinase domain, further facilitating protein kinase cascade (PI3K/AKT/mTOR, ERK/mitogen-activated pathway, etc.) and resulting in cellular proliferation, invasion, migration, and angiogenesis [30]. The enhanced HGF levels were observed to be associated with reduced recurrence-free intervals and less survival outcomes [29]. Various studies revealed that overexpression of c-Met in TNBC is due to the increased copies of a c-Met proto-oncogene [31,32]. In both pre-clinical and clinical trials, it was observed that the combination of overexpression of c-Met proto-oncogene, and loss of p53, led to the development of the claudin-low subtype of TNBC [28].

#### 2.1.6. Programmed Cell Death 1 Ligand (PD-L1)

Programmed cell death 1 ligand 1 (PD-L1) is a transmembrane protein in NK cells, B-cells, activated cytotoxic T cells, and vascular endothelial cells. CD274 encodes this protein and is a checkpoint regulator during the immune response. PD-L1 is a predictive biomarker. The binding of PD-1 with the PD-L1 ligand was observed to inhibit the IL-2 release, T-cell activation, and cell proliferation, thereby inhibiting the functioning of adaptive immune responses [33,34]. It was observed that the PD-1 binding regulates its tolerance to antigens and the expiration of immune responses, thereby restricting autoimmunity in a normal physiological situation. In contrast, in the presence of a tumor microenvironment, the binding serves as a pro-tumorigenic pathway that deactivates T-cells, further facilitating the escape of tumor cells from the immune surveillance. Various studies revealed that the activation of PD-L1 was controlled by multiple signaling pathways like the PI3K/AKT pathway, MAPK signaling pathway, JAK-STAT signaling pathway, aberrant WNT/β—catenin signaling pathway, NF-κβ signaling pathway, and hedgehog signaling pathway. It was observed that 20% of TNBC patients showed expression of PD-L1 [35].

#### 2.1.7. Adenosine 2B Receptor (A2BR)

Adenosine receptors are G-protein coupled receptors that exist as four subtypes, namely Adora1 (A1R), Adora 2a (A2AR), Adora2b (A2BR), and Adora3 (A3R), which are characterized as either pertussis toxin-sensitive (A1R, and A3R), or pertussis toxin-insensitive (A2AR, and A2BR). A2BR is a type of prognostic biomarker. It was observed that cancer growth and progression depend on certain chemical messengers like cytokines, growth factors, and molecules like ATP and adenosine (Ado). Such findings also revealed that in the case of tumor hypoxia, the cell’s metabolic rate gets increased, demanding a high amount of ATP, which then gets metabolized to adenosine (Ado), further facilitating angiogenesis and inflammation, which are considered the two hallmark characteristics of cancer growth. It was observed that A2BR was found to be highly expressed in TNBC. It was further observed that A2BR mediates cAMP signaling, which inhibits the activation of T-cell receptors. This results in cell proliferation, invasion, and the secretion of anti-tumor cytokines like TNF-α and IFN-Υ. Interestingly, it was also revealed that Ado stimulated the expression of VEGF, which leads to enhanced intratumoral blood flow and angiogenesis by mediating purinergic P1 receptors, i.e., A2BR. Likewise, apart from VEGF, the activation of A2BR within the microvasculature also regulates the expression of other angiogenic factors like IL-8 and bFGF as well as results in the proliferation of cells which have an impact on cancer growth, invasion, and migration by inducing neo-vascularization within surrounding areas of cancer [36,37].

#### 2.1.8. CD73

CD73 is a cell-surface glycosylphosphatidylinositol (GPI) that transforms extracellular adenosine monophosphate (AMP) into adenosine and inorganic phosphate [38,39]. It was observed that CD73 plays a role in regulating cancer growth and progression. CD73 is also a prognostic biomarker. On the genetic level, CD73 is considered an ectonucleotide, playing an important role in the purinergic CD39/CD73/adenosine signaling pathway. Further, it was observed that CD73 is responsible for the proliferation, migration, and angiogenesis of TNBC by mediating various signaling pathways, including the EGFR/Akt and VEGF/Akt signaling pathways. In addition to these, CD73 is also responsible for offering resistance to chemotherapy [38]. It was also observed that in TNBC, hypoxia induces the expression of CD73 by activating hypoxia-inducible factor-1α (HIF-1α), promoting epithelial-mesenchymal transition (EMT), cell invasion, and migration. Additionally, it was revealed that CD73-overexpressed TNBC is associated with poor outcomes due to immune evasion, as the adenosine safeguards the cancer cells from adaptive antitumor immune responses [39].

#### 2.1.9. GABA Receptor π Subunit (GABRP)

GABRP is a prognostic biomarker. The GABRP gene encodes the π subunit of the GABA (gamma-aminobutyric acid) A receptor. Various studies found a correlation between the π subunit of the GABA A receptor and basal-like breast cancer subtypes including TNBC. It was revealed that the GABA-π subunit promotes cancer growth in cancer through ERK1/2 signaling. In addition, it was also revealed that breast cancer cell metastases to the brain show a GABAergic phenotype comprising the activation of the GABA A receptor, GABA transporters, and expression of GAD [40]. It was observed that 46–50% of TNBC patients exhibit brain metastasis, which correlates with poor survival. Brain metastasis involves cancer cell invasion, intravasation, and migration to brain cells by bypassing the blood-brain barrier (BBB) [41]. The RT–PCR study further revealed that the metastatic TNBC patients exhibited eight times higher GABRP expression than non–metastatic stage II–IV TNBC patients [42].

#### 2.1.10. G–Protein-Coupled Receptor 161 (GPR161)

G–protein-coupled receptors (GPCRs) are found to be mutated or overexpressed in approximately 20% of all types of cancer, including TNBC, and are considered prognostic biomarkers. GPCRs are heptahelical membrane proteins essential in transducing signals from various ligands. From the genomic profiling, G–protein-coupled receptor 161 (GPR161) was found to be overexpressed in TNBC and is correlated with poor prognosis. GPR161 overexpression was associated with cell growth, proliferation, intracellular accumulation of E-cadherin, cell invasion, migration, and the development of multiacinar structures [43]. From various studies, it was observed that GPR161 knockdown diminishes cellular proliferation. Further, it was revealed that GPR161 forms a complex with multiple scaffold proteins, namely β-arrestin 2 in an ‘agonist-dependent’ manner [44] and Ile Gln motif-containing GTPase Activating Protein 1, and also binds with serine unit of IQGAP1, which overall leads to the activation of mTORC1, which is a sub-unit of the PI3K/AKT/mTOR signaling pathway, promoting cell proliferation and metastasis [45].

#### 2.1.11. G–Protein-Coupled Kisspeptin Receptor (KISS1R)

G–protein-coupled kisspeptin receptor (KISS1R) is another type of GPCR associated with the progression of TNBC and is a prognostic biomarker. KISS1R, also known as GPR54, is a Gα–q/11–coupled GPCR, was found to be overexpressed in TNBC, and is associated with tumor invasion and migration [46]. In TNBC, the overexpression of KISS1R promotes EMT and results in tumor invasion by mediating MAPK and MT1-MMP signaling pathways and activating MMP-9 [47]. Further, KISS1R results in drug resistance by increasing the expression of ERK, AKT, and survivin [48]. It was also observed that the KISS1R pathway includes AXL as its signaling partner, and this was overexpressed in TNBC, depicting poor prognosis in TNBC patients [46].

#### 2.1.12. Intercellular Adhesion Molecule-1 (ICAM-1)

Intercellular adhesion molecule-1 (ICAM-1) is a glycoprotein belonging to the immunoglobulin superfamily. It serves as an adhesion molecule. In addition to this, it elicits metastatic signaling [49]. ICAM-1 is a prognostic-type biomarker of TNBC. It was observed that ICAM-1 was upregulated in various cancers including TNBC. It was revealed that ICAM-1 forms a cross-linking that causes protein phosphorylation, modifications of the cytoskeleton, and the regulation of genes responsible for cell shape and migration [50]. It was observed from various studies that TNBC was associated with metastasis to the lungs, bone, and brain, and it was revealed that ICAM-1 overexpression resulted in lung metastasis of the TNBC cells. It was further observed that endothelial ICAM-1 facilitates the adhesion of leukocyte to endothelium via ICAM-1-LFA1 (lymphocyte function-associated antigen 1) and ICAM-1-Mac1 (macrophage-1 antigen) intercellular interactions, further mediating leukocyte transendothelial migration (TEM). ICAM-1 signaling also sustains the expression of CDK6 and other related pathways related to cell cycle and cell survival [51].

#### 2.1.13. Leptin Receptor

Some studies revealed that the TNBC is also associated with weight gain (obesity), which in turn is associated with the excess secretion of adipokine protein, named Leptin (16kDa), by the adipocytes in response to obesity-related stimuli [52,53]. It was observed that 70–80% of TNBC cases show overexpression of leptin receptors. The leptin receptor is a prognostic-type biomarker of TNBC. It was found that leptin induces the growth and proliferation of cancer and mediates drug resistance [52]. It was observed that the binding of leptin to the leptin receptor facilitates the recruitment of JAK2 kinase, which later leads to the phosphorylation of STAT3 (pSTAT3), activating various downstream signaling pathways (Notch, JAK2, PI3K/AKT/mTOR, and MAPK), genes (Wnt4, ADHFE1, RDH5, etc.), and RBP-JK transcription factor, which is responsible for cell growth, proliferation, migration, and angiogenesis. Leptin binding to the leptin receptor also increases the progression of the cell cycle’s S-phase, apoptosis evasion, and chemoresistance [53].

#### 2.1.14. Monocyte Chemoattractant Protein-1 (MCP-1)

In recent research, it came to light that obesity-related inflammations are also involved in cancer metastasis by producing specific chemokines. The Monocyte chemoattractant protein 1 (MCP-1) is one tumor-promoting chemokine associated with cancer progression. MCP-1 is also a prognostic biomarker. MCP-1 (12kD protein) belongs to the family of the C–C motif chemokine, which binds with the CCR2 receptor, which is a GPCR [54], where it recruits monocytes that later secrete CCL2 chemokine, resulting in tumor proliferation and invasion [55]. Additionally, it was demonstrated that overexpression of MCP-1 is associated with increased accumulation of M2 macrophages and their infiltration into the tumor microenvironment, mediating macrophage-driven angiogenesis [56]. It was observed that MCP-1 overexpression mediates cell invasiveness by activating the p44/42 MAP kinase (MAPK) signaling pathway [54].

#### 2.1.15. Metabotropic Glutamate Receptor-1 (mGluR1)

Metabotropic glutamate receptors (mGluR1–mGluR2) are seven transmembrane domain receptors belonging to GPCRs that facilitate various responses of signaling molecules like chemokines, hormones, neurotransmitters, autocrine, and paracrine factors [57]. It was observed that out of eight mGluRs, mGluR1 and mGluR5 are associated with inducing strong pre-synaptic stimulation [58]. mGluR1 is a prognostic-type biomarker. It was further found that approximately 56% of TNBC patients showed overexpression of mGluR1 [59]. mGluR1, when coupled with Gαq-like protein, activates certain pro-proliferative signalings such as in phospholipase C (PLC), which facilitates the conversion of phosphatidylinositol into IP3 and DAG, activating MAPK and PI3K signaling cascade, which leads to various cellular functions like the regulation of the cell cycle and the activation of pro-survival and antiapoptotic proteins. It was also observed that, in TNBC, mGluR1 triggers the release of pro-inflammatory factors associated with metastasis of TNBC, namely TNF-β, IFN-α, and endothelial cells [60].

#### 2.1.16. MDM2-Binding Protein (MTBP)

MDM2-binding protein (MTBP) is the transcriptional target of MYC oncogene, found to be overexpressed in various cancers including TNBC [61]. It is well known that MYC is a highly preserved oncogenic transcriptional factor that is overexpressed in various cancers and controls oncogenic behavior like increased cell differentiation, proliferation, metastasis, and apoptosis evasion. In addition to the stated activity, MTBP is associated with increased DNA replication [62]. MTBP is also a prognostic biomarker. Moreover, it was found that overexpression of MTBP prevents the self-ubiquitination of Mdm2, which causes Mdm2 stabilization and the enhanced degradation of the tumor suppressor gene, named p53, causing the growth and proliferation of TNBC. Additionally, it was noticed that on metastasis, the expression of MTBP gets downregulated temporarily; however, on getting localized into the metastasized site, the expression of MTBP gets upregulation, resulting in the proliferation of the TNBC cells [61].

#### 2.1.17. Claudin Proteins

Claudins are tight junctional proteins existing between the epithelial cells, creating a barrier for the transport of macromolecules. However, in neoplastic cells, these tight junctions experience structural and functional defects which destroy them [63]. Various studies showed that 66.1% of TNBC cases show increased expression of claudin-4, along with an evident positive correlation with tumor size, nodal status, metastasis, and an expression of Ki-67 [63]. Claudin proteins are also considered prognostic-type biomarkers. Recently, it was found that in addition to claudin-4, claudin-3 and claudin-7 are also regarded as good prognostic factors in TNBC, and this was found relevant through their aberrant immunohistochemical expressions. It was further documented that increased expression of claudin-3 was correlated with the mutation of BRCA1 genes, and this further aids in testing BRCA mutation for TNBC patients [64]. There is another claudin protein named claudin-1 which, unlike the claudins mentioned above, serves as a tumor suppressor in TNBC. It was documented that the resurfacing of claudin-1 on TNBC cells induces apoptosis. From various clinical studies, it was observed that loss of expression of claudin-1 is associated with malignancy, invasiveness, and recurrence of TNBC [65].

#### 2.1.18. Caveolin Proteins

Caveolins (Caveolin-1, 2, and 3) are scaffold proteins composed of cholesterol-enriched microdomains and play an essential role in tumor progression. It was further found that among various caveolins, caveolin-1 (Cav 1) plays a potential role in membrane trafficking, cell invasion and proliferation, cell migration, cell metastasis, and apoptosis, and belongs to the prognostic category of biomarkers. However, it was found that caveolin-1 can function as either a tumor suppressor or promoter, depending on the subtype of cancer in question. Cav-1 acts as an anti-proliferative factor in TNBC by arresting the cell cycle at the G2/M phase, which can be promoted by upregulating specific tumor suppressor genes, namely p21 and p27, and downregulating cyclin D2 [66]. In recent data, it was observed that the loss of normal Cav-1 is linked with the phosphorylation of AKT, TGF-β1, and acceleration of the aggressiveness of TNBC [67].

#### 2.1.19. CCR5

CCR5 (C–C chemokine receptor type 5) is a seven-transmembrane GPCR highly expressed in TNBC patients. CCR5 is also a prognostic biomarker. One cohort study found that approximately 95% of TNBC patients were CCR5+, compared to the percentage of patients positive for CCR5 with other breast cancer subtypes [68]. It was observed that when the promoter region of CCR5 gets methylated, CCR5 protein results in overexpression [69]. It was further observed that overexpression of CCR5 results in increased Ca^2+^ signaling, which facilitates cellular migration in cancer cells. CCR5 also plays an important role in cell growth, proliferation, and the differentiation of immune cells by activating the PI3K signaling pathway, thereby inducing the activation of PDK1 and AKT [68]. Various studies showed that CCR5 overexpression is also positively associated with tumor immune cell infiltration via the activation of effector T-cells and tumor suppressor genes, and repression of YAP1 oncogenic pathways [69].

Recently, it was observed that blocking CCR5 results in anticancer activity. Such a phenomenon was showcased by the emergence of a humanized monoclonal antibody, Leronlimab (PRO 140), and CCR5 antagonist, maraviroc or vicriviroc. It was observed from the preclinical trial that the binding of Leronlimab to human CCR5 leads to the blockage of the CCR5-mediating signaling pathway, thereby preventing TNBC cell invasion [70,71].

Additionally, various in vitro and in vivo studies demonstrated that blocking or knocking down CCL5/CCR5 is harmful to metastatic tumors like TNBC, and thus limits their metastases. In May 2019, Leronlimab (PRO 140) was granted Fast Track Designation by the FDA for its application as a combination therapy with HAART for HIV-infected patients. Recently, Leronlimab has been filed as a drug of choice with the FDA for the treatment of CCR5+ mTNBC patients [72]. The filing was supported by the data from the second patient dosed with Leronlimab (Pro 140) under an emergency investigational new drug (IND) application granted by the FDA in September 2019. It was revealed that the TNBC patients receiving Leronlimab (PRO 140) exhibited no indication of metastases in the lungs and brain during the treatment [73]. In a similar context, phase Ib/II clinical study is ongoing for combining leronlimab with carboplatin (chemotherapy) for the treatment of CCR5+ mTNBC (NCT03838367). The preliminary studies showed an acceptable tolerability and efficacy with an increase in overall survival (OS) and progression-free survival (PFS) [7]. It was observed from the study that the patients who received leronlimab showed a significant 400–660% increase in 12-month PFS, as well as a 570–980% increase in 12-month OS, with a 72% decrease in circulating tumor cells [74].

Moreover, compassionate Use (NCT04313075) and the Basket Study (NCT04504942) were performed to evaluate the safety and efficacy profile of leronlimab at 12 months [75]. In compassionate study (NCT04313075) 2020, leronlimab (PRO 140) was combined with the treatment of physician’s choice (TPC) which included eribulin, gemcitabine, capecitabine, paclitaxel, nab-paclitaxel, vinorelbine, ixabepilone, or carboplatin for the treatment of CCR5+ mTNBC [76]. In the Basket Study (NCT04504942) of 2020, leronlimab (PRO 140) was administered to CCR5+ locally advanced or mTNBC patients. In this study leronlimab (PRO 140) was administered in continuation to the standard-of-care chemotherapy or radiotherapy [77].

In addition, the antibodies ipilimumab (NCT03546686) and tremelimumab (NCT02527434), which target CTLA4, Lacnotuzumab (NCT02435680, targeting CSF1/MCSF), tigatuzumab (NCT01307891, targeting human death receptor 5), utomilumab (NCT02554812, targeting CD137), and LAG525 (NCT03499899, targeting lymphocyte activation gene-3), are being actively analyzed for targeting TMNC (phase II clinical trial) [7].

In addition, the CCR5 antagonist blocks the CCR5 HIV co-receptor, which further leads to decreased in vitro invasion without affecting cell proliferation, and specifically, maraviroc decreases pulmonary metastasis [70,71].

#### 2.1.20. Trop 2

Trophoblast cell surface antigen 2 (Trop-2) is an epithelial membrane surface glycoprotein that plays a pivotal role in cell growth, proliferation, and differentiation, and is found to be overexpressed in TNBC. It is both a prognostic as well as predictive type of biomarker. The overexpression of TNBC is associated with the transcription of various pro-oncogenes like NF-κβ, HOX, etc. The upregulation of Trop-2 was also initiated with the inactivation of TP63/TP53L, ERG, FOXP3, and other transcriptional factors. It was observed that overexpression of Trop-2 knocks out the TACSTD2 gene, further aiding in increased cell growth and proliferation [78]. Apart from regulating transcriptional factors, Trop-2 takes part in Ca^2+^ signaling where it mobilizes calcium into the cells, activating the MAPK, NF-κB, and RAF pathways, and increasing the expression levels of phosphorylated ERK1, ERK2, and FOXM1, thus resulting in enhanced cell proliferation, cell invasion, and metastasis. It was further noticed that the direct interaction of Trop-2 with β-catenin stimulated stem-cell-like properties. Clinically, it was found that antibody-drug conjugates (ADCs) serve as a therapeutic target for Trop-2 [79].

### 2.2. TNBC Biomarkers in the Cytoplasm

#### 2.2.1. PI3K/AKT/mTOR Pathway

The PI3K/AKT/mTOR pathway is the pathway responsible for establishing a balance between two signaling molecules, namely phosphatidylinositol (4,5)—bisphosphate (PIP2) and phosphatidylinositol (3,4,5)—trisphosphate (PIP3), and acting antagonistically with PTEN (phosphatase and tensin homolog) [80]. It was observed that when the growth factors stimulate the signaling pathway, the phosphatidylinositol 4,5-bisphosphate 3- kinase catalytic subunit alpha isoform (PI3KCA) gets activated. The levels of PIP3 get increased, eventually driving the phosphorylation of protein kinase B (AKT) and other downstream functionalities like cell division, differentiation, and survival [81,82]. Further, it was observed that on hyperactivation of the PI3K signaling pathway, various oncogenes (PIK3CA, AKT, and mTOR) get activated, and tumor suppressor genes (PIK3R1, INPP4B, PTEN, TSC1, TSC2, and LKB) get inactivated [83]. The PI3K/AKT/mTOR signaling pathway is a predictive type of biomarker. It was observed that 10% of TNBC patients exhibit PI3KCA mutation while 30–50% of TNBC patients exhibit loss of PTEN expression [82]. The loss-of-function mutation of PTEN includes frameshift mutation and truncated mutation or homozygous deletion, which causes the loss of its functions, i.e., tumor suppression and the hyperactivation of AKT, which furthers lead to cell proliferation, resistance to apoptosis, and the switching of p27 from tumor suppressor to an oncogene [84,85].

#### 2.2.2. Androgen Receptor (AR)

The androgen receptor (AR) belongs to the steroid receptor family and is a nuclear transcription factor. AR is usually found in the cytoplasm and translocates to the nucleus when bound with a ligand. It further gets attached to the androgen-related elements and facilitates cell proliferation [86]. It was observed that AR is overexpressed in 30–35% of TNBC cases [14]. AR is a prognostic biomarker of TNBC. It was further revealed that AR plays an essential role in the progression of TNBC. However, the impact of AR signaling on the prognosis of the TNBC patient remained controversial. It was indicated that being a transcriptional factor, AR controls specific genes associated with particular cell processes, such as stimulating or suppressing cell growth and cell death, etc. [87]. It was further observed that AR overexpression is associated with LAR-subtype TNBC. It was further observed that AR-positive TNBC exhibits decreased Ki-67 index and poor sensitivity to chemotherapy [14].

#### 2.2.3. Aldehyde Dehydrogenase 1 (ALDH1)

Aldehyde dehydrogenase 1 (ALDH1) is a stem-cell-related marker in the cytoplasm of tumor-initiating cells [88]. ALDH1 was found to be overexpressed in TNBC patients and is associated with metastasis (tumor grade) and resistance to chemotherapy (taxane- and epirubicin-based) [89]. ALDH1 is a predictive-type biomarker. It was observed that ALDH1 is associated with cancer stem cells and results in early differentiation [90]. Further, on genomic profiling, it was demonstrated that SMAD4 was the transcription factor of ALDH1. SMAD4 facilitates the TGF-β signaling pathway and regulates genes associated with stemness like Twist1, Snail, and Slug. Thus, it could be inferred that ALDH1 plays a role in cellular differentiation, invasion, tumor development, apoptosis, and immune response [91].

#### 2.2.4. HOX Genes

Abnormal expression of the HOX genes was found to be associated with the growth and proliferation of breast cancer. The HOX genes are considered prognostic biomarkers. The HOX genes are classified into four groups: HOXA, HOXB, HOXC, and HOXD. It was observed that the HOX genes were overexpressed in the primary cancer site with a prominent chance of metastasis. It was revealed that HOXB7 facilitates TNBC progression by activating the TGF-β/SMAD3 signaling pathway through SMAD3 phosphorylation [92]. Moreover, it was observed that HOX genes were regulated by the hypermethylation of the CpGs and epigenetic methylation, which further led to breast cancer tumorigenesis. It was further observed that primary TNBC cells showed three-fold overexpression of HOXB7 compared to normal breast cells [93]. It was also revealed that, contrary to the tumorigenic property of HOXB7, HOXD8 was considered a tumor suppressor gene. It was observed that HOXD8 overexpression diminishes the phosphorylation of AKT and mTOR, which further inactivates the AKT/mTOR signaling pathway and decreases tumor growth and proliferation [94].

#### 2.2.5. Protein Kinase D1 (PKD1)

Protein Kinase D1 (PKD1), belonging to the Ca^2+^/calmodulin-dependent protein kinase (CAMPK) superfamily, is a serine/threonine kinase found to be expressed in almost all tissues. It is found to be overexpressed in cancers, including TNBC. It is a prognostic type of biomarker. It was observed that the activation of PKD1 was mediated in two ways: first was by the phosphorylation of two serine residues (S738/742) located at the activation loop of the catalytic core of protein kinase C (PKC), and the second was through the autophosphorylation of carboxy-terminal of the serine residue (S910). The activation of PKD1 further facilitates various oncogenic activities like cell proliferation, cell survival, migration, and membrane trafficking [95,96].

#### 2.2.6. 6-Phosphofructo-2-Kinase/Fructose-2,6-Biphosphate-4 (PFKFB4)

The most well-known characteristic of cancer cells is the huge production of lactate and pyruvate due to increased glycolysis despite oxygen availability. In this context, PFKFB4 plays an important role in glucose catabolism by regulating glycolytic flux. Moreover, it was observed that PFKFB4 facilitates hostile TME, which promotes the development of tumors in distant sites [97]. PFKFB4 is a prognostic biomarker of TNBC. One of the two primary isoenzymes of the family, namely PFKFB4, was found to be overexpressed in TNBC, and is associated with the regulation of the cell cycle, apoptosis, and autophagy. It was further found that the regulation of PFKFB4 expression was activated by HIF-1, which facilitates the cell to adapt to hypoxia and upregulates the expression of genes responsible for conducting glycolysis. Additionally, PFKFB4 was found to regulate the G1/S phase transition by enhancing the level of CDK6 and phosphorylating Rb. There is some evidence linked with the oncogenic activity of PFKFB4 which states that in TNBC, PFKFB4 regulates cell survival by regulating AKT signaling, the activity of caspase 3/7, and levels of ROS [98].

### 2.3. TNBC Biomarkers in the Nucleus

#### 2.3.1. BRCA Genes

BRCA genes (BRCA1 and BRCA2) are tumor suppressor genes. They are predictive biomarkers of TNBC. They also repair DNA damage, recombine DNA strands, control cell-cycle checkpoints, and regulate apoptotic and transcriptional factors. It was observed that the mutation of BRCA genes leads to the impairment of their functioning as in the impairment of the DNA-repairing and recombination mechanism, disruption in controlling cell-cycle checkpoints, and dysregulation of apoptotic and transcriptional factors, and that such mutations are associated with the progression of breast cancer and ovarian cancer. According to some studies, it was revealed that TNBC shows the mutation of BRCA1 genes (BRCA1Mut), while BRCA2 mutation tends to exhibit similar pathological characteristics as those of normal BRCA genes. It was observed from the epidemiological survey that 5–10% of newly diagnosed TNBC cases in Western countries are associated with BRCA1 (8.5%) and BRCA2 (2.7%) genes (BRCA1/2), and from the current meta-analysis, it was further revealed that such mutation is affiliated with a 40–57% lifetime risk of female TNBC. It was further shown that TNBC patients with BRCA1/2 carriers exhibited a high risk of contralateral breast cancer (≈50%) [99,100].

#### 2.3.2. TP53

One of the important reasons for the failure of TNBC therapy is the lack of identifiable targeted molecular alterations responsible for TNBC progression, therapy resistance, and relapse. The current studies found that ≈80% of TNBC cases showed mutations of the TP53 gene that further led to the production of a mutant p53 protein. Like BRCA genes, TP53 genes are also considered tumor suppressor genes that modulate cell cycle functioning, repair DNA strands and apoptosis, and are predictive biomarkers (69). It was further observed that most TP53 mutation occurs at the DNA-binding domain, resulting in a dysfunctional cell cycle [101]. It was also found that TP53 mutation is associated with the overexpression of CDK7, an essential component of CDK-activating kinase, and plays a pivotal role in cell division and transcription [102].

#### 2.3.3. Activating Transcription Factor 4 (ATF4)

Activating transcription factor 4 (ATF4) belongs to the ATF/CREB family and functions as a transcription factor. It is found to be overexpressed in various tumors, including in TNBC tumors. ATF4 is a prognostic biomarker. It was observed that when a cell experiences stress, like hypoxia, endoplasmic reticulum stress (ERS), or nutrient deprivation, the integrated stress response (ISR) gets activated, and this helps in preserving homeostasis. Further, the ISR activation results in the reduction of global protein synthesis through the phosphorylation of eIF2α (eukaryotic translation initiation factor 2 alpha), which ultimately leads to the activation of ATF4, which controls the fate of the cancer cells by regulating cell growth, proliferation, invasion, migration, autophagy, and resistance to chemotherapy. It was further confirmed that the phosphorylation of eIF2α was initiated via double-stranded RNA-dependent protein kinase (PKR, EIF2AK2), endoplasmic reticulum kinase (PERK, EIF2AK3), general control nonderepressible 2 kinases (GCN2, EIF2AK4), and heme-regulated inhibitor (HRI, EIF2AK1) [103]. It was observed that in TNBC, overexpression of ATF4 was correlated with poor OS after diagnosis (approximately 37 months) [104]. Genetic profiling further revealed that the overexpression of ATF4 was associated with the canonical SMAD-dependent TGF-β pathway. The study depicted that the depletion of ATF4 further reduced the activity of TGF-β and diminished the expression of the SMAD2/3/4 pathway, indicating the existence of a feedback loop between ATF4 and the TGF-β pathway. Such findings further demonstrated that TGF-β and SMAD2/3/4 constitute the upstream signaling pathway of ATF4, regulating the positive feedback loop of the TGF-β pathway associated with TNBC aggressiveness [103].

#### 2.3.4. ETS Translocation Variant4 (ETV4)

ETS translocation variant (ETV4) is a transcription factor belonging to the PEA3 subfamily of ETS (E–26). It was observed that in cancer cells, ETV4 is expressed at a higher level compared to normal cells. ETV4 also acts as an oncogenic protein that can enhance cancer growth, progression, and metastasis. Further analysis showed that EGFR induces nuclear translocation of ETV4 by activating matrix metalloproteinase (MMP)–9 and 14. It was observed that 57% of TNBC cases exhibit overexpression of ETV4 proteins. ETV4 is a prognostic biomarker. ETV4 protein overexpression was also associated with lymph nodes and lymphovascular invasion [105]. Additionally, ETV4 mediates the expression of MMP13, which plays an important role in proliferation, invasion, and migration [106].

#### 2.3.5. Forkhead Box M1 (FOXM1)

Forkhead Box M1 (FOXM1) belongs to the family of fork-head/winged-helix proteins associated with various biological processes like cell proliferation, differentiation, DNA damage repair, and angiogenesis [107]. Various studies revealed that FOXM1 was found to be overexpressed in cancers including TNBC. Notably, 85% of TNBC patients show overexpression of FOXM1, which serves as a prognostic biomarker. In TNBC, it was observed that FOXM1 promotes cell progression and invasion by direct binding with eEF2K [108]. Wei et al., 2015 further revealed that FOXM1 regulates EMT by activating the SNAIL gene. Studies also demonstrated that FOXM1 activation or overexpression also excises other biological processes in TNBC via reprogramming energy metabolism, inflammation, apoptosis evasion, promoting genomic instability, and enabling replicative mortality [109].

#### 2.3.6. Glucocorticoids

In recent times, steroid receptors have emerged as potential prognostic and predictive-type biomarkers in TNBC. Among various steroid receptors, glucocorticoid receptors (GRs) are overexpressed in TNBC. It was observed that the binding of GC to GRs facilitates the translocation of GC to the nucleus, where it undergoes dimerization and enhances the transcription of GC-inducible genes, which results in anti-apoptotic activity and multi-drug resistance [110]. Preclinical studies revealed that GRs’ antiproliferative effect was mediated via BRCA1, where their activity resulted in the phosphorylation of downstream signaling pathways like MAPK. However, some pieces of evidence indicate that the long-term activity of GRs diminishes the expression of BRCA1, whereas the accumulation of free GRs enhances the expression of BRCA1. Such statements require evidence circumventing the specific mechanisms of GRs so that they can be employed as applicable proteomic biomarkers in the therapy of TNBC [111]. Recent studies have revealed that an increased mortality rate was observed in TNBC patients that exhibited overexpression of GRs because GR overexpression can activate the oncogenes, leading to the suppression of the tumor suppressor gene and preventing apoptosis and resulting in unfavorable clinical outcomes. Further, it was observed that in addition to MDR, GR overexpression is also associated with increased recurrence [112].

### 2.4. TNBC Biomarkers in the Blood

#### 2.4.1. Vascular Endothelial Growth Factor (VEGF)

Cancer cells require oxygen and nutrients for their growth and angiogenesis, i.e., the formation of new blood vessels aids in providing these substrates. The critical facilitator of angiogenesis is vascular endothelial growth factor (VEGF), mostly induced by hypoxia. Hence, VEGF can be considered an appealing target for developing anticancer therapeutics and is a prognostic biomarker of TNBC [18]. VEGFs were found to be overexpressed in 30–60% of TNBC patients [14]. It was also observed that VEGFR gets activated in the presence of a mutated p53 gene. The activated VEGFR further stimulates the JAK2/STAT3 signaling pathway and increases proliferation, migration, angiogenesis, and chemoresistance [113]. It was depicted that in cancer, VEGF gets glycosylated and further binds with VEGFR, stimulates the process of angiogenesis, and increases the permeability of neighboring blood vessels and lymphatics, ultimately resulting in increased cancer metastasis [14].

#### 2.4.2. Interleukin-8 (IL-8)

The progression of TNBC needs the simultaneous expression of interleukin 8 (IL-8). In TNBC, IL-8 acts as a predictive biomarker. From the xenograft animal model, it was observed that inhibiting the expression of IL-8 facilitated the suppression of cell progression, colony formation, migration, etc. [114]. IL-8 is a pro-inflammatory multifunctional chemokine that binds with the chemokine receptors, namely CXCR1 and CXCR2, and as a result the various signaling pathways like mitogen-activated protein kinase (MAPK) and Akt get activated. Additionally, it was observed that the production of IL-8 can be regulated via various factors like lipopolysaccharide (LPS), tumor necrosis factor-α (TNF-α), and IL-1β. In cancer, IL-8 overexpression is associated with the induction of cyclin D1 and B1, resulting in increased tumor progression, angiogenesis, and cell invasion and migration [114,115]. In TNBC, IL-8 production is associated with hypoxic conditions and aids in recruiting mesenchymal stem cells (MSCs) to the primary site of TNBC, creating a microenvironment around the tumor. Such aberrant physiological conditions increase multi-drug resistance (MDR) and metastatic risk [10].

Although we have described the role of each biomarker in the progression of TNBC, it was recently observed that scientists are endorsing a combination of biomarkers for better and more efficient results. In such a scenario, the specificity and the sensitivity of individual biomarkers are kept optimum so that they can result in a better prognosis or an effective diagnosis. Hence, before selecting a biomarker or combination of biomarkers, the family history of the concerned patient, and their lifestyle, should be considered. Additionally, specific ideal characteristics of biomarkers recorded include the expression of biomarkers in the early stage of the disease and the ability to discriminate the diseased population from the healthy population. All these factors and criteria result in effective therapy and diagnosis [116].

In Table 1, we have listed various clinical studies that have been completed or are ongoing involving the participation of biomarkers in the treatment of TNBC.

## 3. Targeted Therapies Based on Biomarker Appraisal

Conventional chemotherapy remains the backbone of TNBC therapy, despite the emergence of various biomarkers. However, the hardship faced by researchers does lead to the development of specific targeted therapies for treating TNBC, as discussed below and shown in Figure 2.

### 3.1. Signaling Pathway Inhibition

As discussed above, cancer cells exhibit highly activated signaling pathways like EGFR, VEGFR, and their downstream pathways such as PI3K/Akt/mTOR. These overexpressed signaling pathways accelerate tumor growth, progression, migration, and angiogenesis. Hence, it was inferred that the inhibition of these overexpressed signaling pathways provides a potential platform for treating TNBC. The targeted therapeutic strategies for TNBC that were evaluated in pre-clinical and clinical studies are overviewed in Table 2.

#### 3.1.1. Inhibition of EGFR Signaling Pathway

EGFR was overexpressed in TNBC and is associated with tumor growth and progression [117]. It was observed that monoclonal antibodies such as cetuximab and panitumumab and inhibitors of a tyrosine kinase such as gefitinib, erlotinib, etc., are used for targeted overexpressed EGFR. Monoclonal antibodies and tyrosine kinase inhibitors are already being approved to treat colorectal and lung cancer [118]. Recently, they have also been subjected to clinical trials for the treatment of TNBC. In the TBCRC 001 study, half of the TNBC patients were administered cetuximab alone, while the other half were given cetuximab in addition to carboplatin.

Similarly, in another study, NCT00463788, metastatic TNBC patients were administered either cisplatin alone or cisplatin in addition to cetuximab. From this study, it was observed that patients receiving cisplatin alone showed a 10% objective response rate (ORR). In contrast, patients receiving cisplatin in combination with carboplatin showed a 20% accurate response rate, in addition to more prolonged progression-free survival (PFS) [119]. The MD Anderson Cancer Center performed a study that revealed that erlotinib, an inhibitor of EGFR tyrosine kinase, enhanced the expression of E-cadherin and decreased the expression of vimentin, facilitating the reversal of mesenchymal phenotype to epithelial phenotype and ultimately resulting in the inhibition of TNBC growth and progression [120]. Similarly, another study was performed with erlotinib in the SUM149 xenograft mouse model which showed that erlotinib both prevented the growth of TNBC as well as inhibited its metastasis [121]. The above findings thus suggested that EGFR targeting could provide a platform for a potential therapeutic approach against TNBC, as targeting EGFR modulated the EMT phenomenon, reduced the metastasis, and inhibited tumor growth.

#### 3.1.2. Inhibition of the PI3K/Akt/mTOR Signaling Pathway

TNBC patients (23.7%) showed mutations of the PI3KCA gene, the pivotal gene encoding the catalytic subunit of the PI3K/Akt/mTOR signaling pathway [122]. Thus, the inhibition of the PI3K/Akt/mTOR signaling pathway provides a therapeutic approach against TNBC. BKM120, the PI3K inhibitor, exhibited a significant inhibition of tumor growth, with 84% tumor-growth inhibition [123]. Lin et al., 2020 used rapamycin, an inhibitor of mTORC1, to treat metastatic TNBC [124]. In phase II clinical trials (NCT02162719), TNBC patients received paclitaxel i.v. with or without Ipatasertib, an inhibitor of the Akt pathway. It was observed that TNBC patients receiving the combination exhibited an increased PFS of 6.2 months, as compared to those receiving only paclitaxel, who exhibited a PFS of 4.9 months [125]. Thus, targeting the PI3K/Akt/mTOR pathway has emerged as a potential therapeutic strategy for treating TNBC.

#### 3.1.3. Inhibition of VEGFR

Angiogenesis is one of the hallmarks of TNBC and is correlated with the overexpression of VEGFR (Vasculo-endothelial growth factor receptor). It was revealed that high VEGFR content is associated with poorer relapse-free survival (RFS) and overall survival (OS) [126]. In the RIBBON-2 phase III trial, Bevacizumab, a humanized antibody binding to VEGF-A, was employed as a second-line treatment against TNBC. It was found that Bevacizumab prevents the binding of VEGF with its receptor, VEGFR. Additionally, Bevacizumab exhibited improved PFS (6 months), OS, and ORR in TNBC [127]. Additionally, in metastatic TNBC, Bevacizumab was employed as a first-line treatment with a 49% enhanced response rate. In the Geparquinto Study (GBG 44), 2013, bevacizumab was administered to TNBC patients in combination with anthracycline-taxane chemotherapy, and these patients exhibited an improved pCR rate of up to 39.3%, as compared to 27.9% in the case of patients with anthracycline-taxane chemotherapy alone [128]. Another inhibitor of VEGFR, Apatinib, was employed in a multicentre phase II clinical trial (NCT01176669) to treat metastatic TNBC. The results revealed that Apatinib showed an improved ORR of 10.7%, PFS of 3.3 months, OS of 10.6 months, and a clinical benefit rate of 25% [129]. The above revelation indicated that angiogenesis inhibitors might provide a platform for building a novel therapeutic strategy against TNBC.

### 3.2. Immune Checkpoints Inhibition

Researchers are showing immense interest in cancer immunotherapy, specifically immunotherapy based on immune checkpoints such as programmed cell death 1 (PD-1), programmed death ligand 1 (PDL1), and cytotoxic T-lymphocyte-associated protein 4 (CTLA4).

The FDA has approved various immune checkpoint inhibitors like nivolumab, ipilimumab, and atezolizumab. All of these approved immune checkpoint inhibitors are humanized antibodies, showing significant benefits in cancer treatment [130]. In general, it has been observed that cancer is not immunologically active. However, the TNBC subtype was found to be responsive to immunotherapy due to the increased occurrence of tumor-infiltrating lymphocytes (TILs) [131]. In KEYNOTE-522 clinical trials (NCT03036488), pembrolizumab was employed in combination with chemotherapy to treat early TNBC. The combination showed a higher pCR rate in the first and second interim, mostly in the early TNBC patients with positive PDL1. Clinical studies also showed a relation between the overexpression of PDL1 and the efficiency of immune checkpoint inhibitors in patients suffering from metastatic TNBC. Recently, two companies provided approved antibody-based diagnostics for the expression of PDL1. One was PDL1 IHC 22C3, which Agilent Technologies developed, and the other was Ventana PDL1 (SP142), developed by Roche Diagnostics. PDL1 IHC 22C3 was employed to screen TNBC patients for pembrolizumab treatment, using 10 combined positive scores (CPS) as cutoffs. In contrast, Ventana PDL1 (SP142) was employed for screening metastatic TNBC patients for atezolizumab therapy, using a cutoff of 1% immune cell score (ICC) [132]. In this context, the phase III KEYNOTE-355 trial (NCT02819518) was performed, where patients with metastatic TNBC were administered a combination of pembrolizumab and chemotherapy or a combination of placebo and chemotherapy. In the study, PDL1 expression was assessed via PDL1 IHC 22C3 assay and characterized by a combined positive score (CPS). It was observed that patients with 10 CPS or more exhibited prolonged median PFS in the case of a combination of pembrolizumab plus chemotherapy, as compared to patients who had had placebos plus chemotherapy [133]. Similarly, in the Impassion130 clinical study (NCT02425891), patients with metastatic TNBC received a combination of atezolizumab and nab-paclitaxel or a combination of placebo and nab-paclitaxel. In this study, a PDL1 (SP142) immunohistochemical assay was employed for assessing the expression of PDL1 on tumor-infiltrating immune cells. It was observed that the patients who had had the atezolizumab-and-nab-paclitaxel combination showed higher PFS as compared to those who had had the placebo plus the nab-paclitaxel [134].

### 3.3. Inhibition of Poly (ADP-Ribose) Polymerase (PARP) Enzymes

It was acknowledged that in cells functional for BRCA1 and BRCA2, the DSBs are repaired via homologous recombination (HR). Hence, it was indicated that cancers with BRAC–mutation are more sensitive to PARP inhibitors due to the loss of both PARP repair and HR repair, a situation known as “synthetic lethality”. As mentioned earlier, approximately 80% of TNBC patients show BRCA1 mutation. Thus, it was inferred that PARP inhibitors could treat TNBC with BRCA1 mutation. Until now, four PARP inhibitors have been approved by the FDA for cancer treatment, namely niraparib, olaparib, rucaparib, and talazoparib, and out of these four, olaparib, and talazoparib have been approved for the treatment of BRCA-mutated metastatic breast cancer. Various inhibitors of PARP have been evaluated in a clinical study for the treatment of TNBC. A phase I clinical study (NCT01445418) was performed where olaparib was administered in combination with carboplatin to metastatic TNBC patients either with no BRCA mutation (cohort I) or with BRCA-mutation with BRCAPro scores of <10% (Cohort II). The ORR was 22%, with the patients having a complete response. Another novel PARP inhibitor, Veliparib, showing favorable toxicity, has not yet been approved by the FDA but is under extensive studies in combination with chemotherapeutics. In a phase I clinical trial, veliparib was combined with cisplatin and vinorelbine to treat BRCA1/2-mutated TNBC. The patients receiving the combination showed 73% ORR. Recently, a phase III clinical trial (NCT02032277) was assessed, where veliparib was combined with paclitaxel and carboplatin as neoadjuvant chemotherapy for the treatment of TNBC [134].

### 3.4. Inhibition of Cell Cycle

The cell cycle comprises four phases, namely the resting stage (G1 phase), synthesis phase (S phase: where DNA replication takes place), cell growth phase (G2 phase: where the cell grows and prepares itself for division), and mitosis phase (M phase: where cell division takes place). Cyclin-dependent kinases (CDKs) are protein kinases that, on binding with the cyclins, promote the cell cycle’s progression [135]. It was observed that cancer associated with dysregulated CDKs facilitated unscheduled proliferation [136]. Hence, it was indicated that CDK inhibitors could provide an approach employed against TNBC. In this context, it was found that the inhibitors of CDK4/6 block the transition of the S phase from the G1 phase by dephosphorylating the retinoblastoma tumor suppressor protein (Rb), thereby inhibiting cell proliferation. The various FDA-approved CDK4/6 inhibitors used are Abemaciclib, Palbociclib, and Ribociclib. In modern times, combination therapy is urging the hype for the treatment of TNBC, and from extensive research, it has been revealed that a combination of drugs, such as those in chemotherapy, PI3K inhibitors, immune checkpoint inhibitors, and CDK4/6 inhibitors, not only provide increased anticancer activity but also overcome multi-drug resistance in TNBC [137]. One of the studies found that the combination of PI3K and CDK4/6 inhibitors provided a synergistic effect and generated immunogenic apoptosis in TNBC cells [138]. Various clinical trials are ongoing regarding the combination of CDK4/6 with other agents including anti-androgen, anti-PDL1 antibodies, and chemotherapeutics for treating TNBC.

### 3.5. Inhibition of Epigenetic Modifications

Epigenetic modifications result in chromosome changes without altering the DNA sequences, thus specifying stable, heritable changes [139]. Recently, epigenetic modifications, like DNA methylation, histone deacetylation, etc., have played an essential role in cancer growth and progression [139]. In DNA methylation, the binding of transcription factors gets inhibited, and in histone modification, the chromatin undergoes phosphorylation, acetylation, and ubiquitinoylation [140]. Thus, the inhibition of DNA methylation and deacetylation of histone may be used as a targeted therapeutic approach against TNBC.

#### 3.5.1. Inhibition of DNMT

DNA methylation adds a methyl group to the cytosine ring (5′ position) in CpG dinucleotides. It was observed that the mutation of BRCA1 was inhibited in TNBC by the prevention of hypermethylation [141]. As observed in a pre-clinical study, combining PARP inhibitors with DNMT (DNA methylation) inhibitors like 5-azacytidine and decitabine increases the efficiency of PARP inhibitors. It inhibits the growth and progression of BRCA1-overexpressed TNBC [142]. Although DNMT inhibitors are in clinical trials for treating TNBC, the USFDA has already approved DNMT inhibitors for treating other cancers like myeloid malignancies, etc. [143].

#### 3.5.2. Inhibition of HDAC

Histone deacetylase (HDAC) causes the deacetylation of histone proteins, leading to the condensation of chromatin and suppressing gene transcription. It was observed that negative regulation of the tumor suppressor gene leads to tumor growth, invasion, proliferation, migration, and angiogenesis [144]. In contrast, the inhibitors of HDAC reverse the suppression of gene expression via histone hyperacetylation and chromatin relaxation. Additionally, HDAC inhibitors induce cell apoptosis, inhibiting cell invasion, proliferation, migration, and angiogenesis. Sulaiman et al., 2018 demonstrated that TNBC showed overexpression of mTORC1 and HDAC as compared to luminal breast cancer, and that the combination of mTORC1 and HDAC inhibitors provided a synergistic activity against TNBC [145]. In another study, it was revealed that the HDAC inhibitor suberoylanilide hydroxamic acid (SAHA) increases the anticancer activity of olaparib, a PARP inhibitor in TNBC, by regulating the homologous recombination repair’s (HRR) expression [146].
cancers-15-02661-t002_Table 2Table 2Overview of the potential targets along with their therapeutic strategies for TNBC.TargetsDrugsPhaseOutcomeRefs.PARPOlaparibI/II/III✓The TNBC patients receiving olaparib showed increased OS, DDFS, and IDFS (OlympiA trial)[147]VeliparibII✓Veliparib, in combination with carboplatin, exhibited an increased rate of pCR as compared to standard therapy (I-SPY 2 trial)[148]IniparibII✓Patients receiving Iniparib showed a higher HRD-LOH score in comparison to those receiving standard therapy (PrECOG 0105 trial)[149]Immune checkpointsPDL1: PembrolizumabFDA-Approved✓Patients receiving a combination of pembrolizumab plus chemotherapy showed an increased rate of pCR; this generally occurred in TNBC patients with overexpressed PDL1 (KEYNOTE-522 trial)✓TNBC patients with PDL1 overexpression receiving pembrolizumab plus chemotherapy exhibited prolonged PFS compared to those receiving chemotherapy alone (KEYNOTE-355 trial)[133,150]PDL1: Atezolizumab, DurvalumabII/III✓PDL1-positive TNBC patients receiving Atezolizumab in combination with nab-paclitaxel showed prolonged PFS (IMpassion130 trial)✓Patients with overexpressed PDL1, and increased stromal TILs, showed an increased rate of pCR for both the durvalumab and placebo group (GeparNuevo study)[134]Signaling pathwaysEGFR: CetuximabII✓Patients treated with cetuximab and cisplatin showed prolonged PFS, with 20% ORR, as compared to those treated with only cisplatin, who showed only 10% ORR[132]EGFR: ErlotinibPre-clinical✓Erlotinib prevented the growth of cancer and inhibited metastasis. Additionally, it facilitated a reversal of phenotype from mesenchymal to epithelialPI3K: BKM120Pre-clinical✓BKM120 exhibited a significant reduction of cancer growth in PDX models[123]Akt: IpatasertibII✓The ipatasertib group showed an increased median PFS of 6.2 months compared to the placebo group, who showed a PFS of 4.9 months only[125]AngiogenesisVEGF: BevacizumabII/III✓TNBC patients treated with Bevacizumab showed an enhanced median PFS of 6 months, median OS of 17.9 months, and ORR of 41%, as compared to those treated with chemotherapy treatment, for whom the median PFS was 2.7 months, median OS was 12.6 months, and ORR was 18% (RIBBON-2 trial)✓Administration of Anthracycline—taxane—containing chemotherapy in combination with Bevacizumab exhibited an enhanced rate of pCR (39.3%), as compared to anthracycline—taxane containing chemotherapy alone (27.9%) (GeparQuinto trial)[127,128]
VEGFR: ApatinibII-The trial containing Apatinib showed improved ORR, clinical benefit rate, median PFS, and OS, as compared to the trial with only chemotherapy[132]Epigenetic modificationDNMT: 5- Azacytidine/AZA, Decitabine/DACPre-clinical-A combination of PARP inhibitor and AZA/DAC demonstrated an enhanced efficacy of PARP, which resulted in an increased inhibition of tumor growth, as compared to the results of free drugs[142]HDAC: Suberoylanilide hydroxamic acid (SAHA), Entinostat (ENT)Pre-clinical-The expression of ER-α was reduced by ENT, which also conserved the sensitization of breast cancer cells towards the inhibitor of aromatase, Letrozole-The combination of SAHA and an inhibitor of PARP, Olaparib, regulated the expression of homologous recombination repair (HRR)-related genes and hampered the DNA-repairing mechanism, thereby enhancing the anticancer activity of olaparib[151]Cell cycleCDK4/6: PalbociclibI/II✓Phase I/II clinical trials accompanying the inhibitors of CDK4/6 for TNBC are ongoing[152]CHK1: MK-8776Pre-clinical✓The inhibitor of CHK1 delayed the repairing of radiation-induced DNA damage, which resulted in inhibiting the survival of TNBC cells[153]


## 4. TNBC Biomarkers in Cancer Nanotherapeutics

Drug delivery refers to delivering pharmaceuticals, small molecules, genes, and biomolecules to a diseased site (cell or organ), facilitating desired therapeutic effects and minimizing side effects [154]. However, it was observed that 80% of clinical drugs fail to produce the desired therapeutic efficacy [155] as they suffer from insufficient bioavailability due to poor water solubility, permeability, and biological barriers. Thus, it was demonstrated that therapeutic performance does not depend merely on the activity of the administered moieties, but also depends on their bioavailability at the target site. Furthermore, it has been observed that conventional drug delivery systems demonstrate severe constraints like non-controlled drug release, non-targeting delivery, systemic side effects, increased and frequent dosing, and poor bioavailability [156].

Various encouraging drug delivery strategies have been evolved to overcome the failure of conventional drug delivery systems, and these include drug optimization, drug modifications, microenvironment modification, and the emergence of novel drug delivery systems. In drug optimization, structure–tissue exposure/selectivity activity relationship (STAR) is employed, improving drug optimization by classifying the drug candidates based on potency, tissue exposure, selectivity, required dose for balancing clinical efficacy, and toxicity. Such an approach overcomes the gaps caused by the structure–activity relationship (SAR), which only indicates potency/specificity. It overlooks tissue exposure/selectivity in disease/normal tissues, thereby misleading the selection criteria in the drug candidate selection and the impact of these criteria on the balance between efficacy and toxicity [155]. In drug modification, the structure of the drug in question gets altered by changing its orientation, nature, or type of functional groups, amino acids, or nucleic acid backbones, thereby improving its pharmacokinetic attributes. Additionally, the drug can be conjugated with known moieties and targeting ligands, thereby increasing its targetability and therapeutic efficacy and improving the drug release profile. Such a strategy aims to modulate the interaction of the drug and the tissues or cells and control the navigation of the drug from its administration to the desired therapeutic activity. In microenvironment modification, the host environment gets altered, and this significantly changes the mechanistic approach of the drug at the site of action, such as by applying pH modifiers, permeation enhancers, protease inhibitors, enzyme inhibitors, etc. Such microenvironment modifications aid in navigating biological barriers [157]. Lastly, pharmaceutical companies are investing more in developing novel drug delivery systems that exhibit excellent therapeutic performance, flexible drug release profile as per the desired disease, clinical efficacy, prolonged product life, increased targetability, reduced dose frequency, and dose-dependent side effects [156].

Nanoparticles have shown evidence that showcases their efficacy in bypassing the limitations of the conventional drug delivery system, through methods such as biodistribution and intracellular trafficking via site-specific targeting. With this realization, the US National Science and Technology Council (NSTC) launched an initiative named National Nanotechnology Initiative (NNI) in 2000 that outlined the efforts to improve therapeutic research through the emergence of nanotechnology [158].

In this section, we discuss the involvement of a novel drug delivery system, namely one using nanoparticles, in treating TNBC, as well as its targeting approach, the different types of nanoparticles, and the challenges these nanoparticles face, along with the strategies employed for their bypassing.

### 4.1. Significance of Nanotherapeutics in TNBC Therapy

TNBC is the most aggressive subtype of breast cancer, with a lack of expression of estrogen, progesterone, and human epidermal receptor-2 receptors. In addition to these, TNBC exhibits heterogeneous genomic profiling with six subtypes, each with different characteristics, and exhibits rapid metastasis to local lymph nodes, brain, lungs, and bones. Such attributes of TNBC make its treatment regimen challenging in therapeutics, as TNBC is insensitive to standard endocrine therapy. Only chemotherapeutics like anthracycline and taxane-based chemotherapy seemed effective against TNBC [6]. However, it was found that these chemotherapeutics also affect healthy cells in addition to cancer cells, thereby causing adverse effects like bone marrow suppression, alopecia, nausea, vomiting, thrombocytopenia, cardiotoxicity (for doxorubicin, specifically), and pulmonary edema (for cisplatin, specifically) [159]. Moreover, chemotherapy causes drug resistance due to the overexpression of the P-gp efflux pump within the cancer cells and the immune escape of tumor cells, and this restricts their therapeutic effects against TNBC [160]. Hence, to bypass such non-targeted effects of chemotherapy, novel molecular targets have emerged and are delivered by nanotechnology-based drug delivery systems.

Past research has revealed that nanoformulations selectively target cancer sites, eliminating the accumulation of chemotherapeutics in healthy cells. Nanotechnology has gained worldwide attention in the field of cancer treatment and diagnosis. In cancer diagnosis, NPs are employed for the identification of cancer biomarkers. Due to the increased surface area to volume ratio of NPs, their surface can be densely covered by antibodies, peptides, aptamers, small molecules, etc., which can bind with the targeted cancer molecules. Thus, binding the functionalized NPs with the specific ligands has been inferred to establish a multivalent effect, further enhancing the assay’s specificity and sensitivity [161]. NPs are also developed as nano-biosensors, detecting multiple protein biomarkers in seconds. Such nano-assisted technology is also associated with imaging applications, helping identify cancer early and more accurately [162]. In cancer therapy and management, NPs aid in targeting the chemotherapeutics specifically for cancerous cells without hampering healthy cells. Their large surface-area-to-volume ratio helps assemble biomolecules over NPs, thereby improving specificity via active targeting and efficacy of the targeted NPs, and reducing the off-target side effects of the chemotherapeutics. Moreover, their small size helps them penetrate the leaky vasculature generated by tumor-induced angiogenesis, thereby avoiding the normal blood vessels and increasing the passive targeting ability of NPs [163]. Hence, it has been inferred that in the treatment field, NPs exhibit distinctive physicochemical characteristics like small particle size, increased surface-to-volume ratio, improved entrapment efficiency, increased adhesion to the tumor microenvironment, controlled and precise drug release, minimum systemic toxicity, and safe biological elimination [162].

While focusing on the mechanistic approach of NPs in diagnosis and therapy, it was revealed from various studies that cancer cells perform intracellular uptake of nanoparticles by three mechanisms, namely passive pathway, active pathway, and triggered targeting. In passive targeting, the cancer cells facilitate intracellular uptake via the enhanced permeability and retention (EPR) effect. In the EPR effect, the nanoparticles undergo intracellular accumulation due to leaky vasculature and a dysfunctional lymphatic system. In an active pathway, the nanoparticles mediate surface modification or functionalization with various molecules like ligands, peptides, antibodies, etc., which are either deregulated or overexpressed in cancer cells. This ligand-binding phenomenon triggers receptor-mediated endocytosis, leading to cellular internalization and specific targeting. Finally, in triggered targeting, the release of drugs from the nanoparticles is triggered upon exposure to distinct external stimuli, and this results in the desired localization of drugs within the target site. The external stimuli can be temperature or pH fluctuation, electric application, magnetic fields, ultrasound, light, etc. However, such triggering systems are challenging to develop and often lead to the unsolicited release of drugs due to the modulations experienced by the tumor microenvironment [164].

The tumor microenvironment (TME) is also gaining much attention in treating cancer cells by using nanoparticles. It has been observed that the TME comprises cellular and structural components like the fibroblast, extracellular matrix, immune cells, and vasculature, which surround cancer sites and help in the growth of cancer cells and in their metastasis. It was further found that the existence of the TME limits the delivery of chemotherapy to the cancer site, thereby leading to the failure of the therapy. Recent treatments like antiangiogenic therapy and immunostimulatory therapy have shown limited success despite demonstrating encouraging pre-clinical results. Such limitations are due to the lack of drug penetration into the necrotic tumor core, non-specific delivery, rapid elimination from serum, and dose-depended toxicity. All these problems were further resolved in other studies by applying nanoparticles that targeted the TME vasculature, ECM, and immune response [165]. Usually, while targeting the TME, various pathophysiological conditions of TME are taken care of, including enzymatic activity, hypoxia, oxidative stress, high interstitial fluid pressure, levels of amino acids, functional proteins, levels of macrophages, lymphocytes, and neutrophils. Pre-clinically, it was observed that NPs target the TME by involving pegylated, stimuli-responsive, and dual-functional nanoparticles. More specific strategies involve site-specific attachment of PEG linkage, surface-charge reversal, decrease in particle size, hyperthermia-induced generation of CO_2_, and response to internal stimuli like pH, temperature, and external stimuli like a magnetic field, light, ultrasound, etc. [166]. Various studies showed that encapsulating cytokines, siRNA, and cytotoxic drugs within nanoparticles induces immune stimulation, which can either kill or modify the tumor-associated macrophages, which are an important component of TME. Additionally, nanoparticles can target distinct immune subpopulations like T cells, NK cells, and DCs through surface functionalization [165]. Besides this, the upsurging understanding and knowledge of targeting the TME using nanoparticles are paving the way for the fabrication of a combined strategy involving therapeutics and diagnostics, commonly known as nanotheranostics.

### 4.2. Correlation of Nanotherapeutics and TNBC Biomarkers

Nanoformulations have benefitted from the enormous amount of information offered by biomarker screening in TNBC. These molecular targets have been employed to fabricate alternative approaches for precision therapy using multi-functional nanoformulations, which can enhance the detection limits of analytes for facilitating more selective treatment through the direct delivery of drugs into cancer cells while sparing healthy cells. Various functionalized nanoparticles have been developed for the treatment of TNBC such as polymeric nanoparticles, polymeric micelles, liposomes, nanoemulsions, solid lipid nanoparticles, nanostructured lipid carriers, dendrimers, metallic nanoparticles, exosomes, quantum dots, inorganic nanoparticles like gold nanoparticles, etc. [167].

#### 4.2.1. Inorganic Nanoparticles

Inorganic nanoparticles like silver nanoparticles (AgNPs), gold nanoparticles (AuNPs), mesoporous silica nanoparticles, and carbon-based nanoparticles are also employed as suitable nanoformulations in cancer diagnostics and treatment. It was observed that AuNPs show characteristic chemical, optical, physical, and electronic features that aid in assisting functionalization to access tumors as well as load high drug doses. In addition to these, AuNPs also exhibit biocompatibility. Carbon-based NPs, specifically carbon nanotubes (CNTs), are also employed as delivery systems because of their capacity to enter the cells via a ‘needle-like penetration’ technique, nano-size, distinctive structure and morphology, physical features, increased loading efficacy, controlled drug release profile, and multifunctional capacity [167].

Webb et al., 2017 engineered multi-branched gold nanoantennas (MGNs) for establishing their theranostic ability against TNBC. The MGNs were functionalized with anti-PDL1 antibodies-DTNB (dithio-bis-(2-nitrobenzoic acid)), and anti-EGFR antibodies—pMBA (para mercaptobenzoic acid) for targeting PDL1, and EGFR, respectively. From surface-enhanced Raman scattering (SERS) imaging, it was revealed that an MGN undergoes cellular localization through receptor-mediated endocytosis and surface binding, which mediated targeted diagnosis for both PD-L1 and EGFR [168]. Harmon et al., 2017 developed iron oxide nanoparticles (IONP), functionalized with LPrA2, an antagonist of leptin, for the treatment of TNBC with an overexpressed leptin receptor. These nanoparticles further encapsulated various chemotherapeutics like cisplatin, doxorubicin, cyclophosphamide, and paclitaxel. It was observed from the study that the prepared IONP-LPrA2 reduced the levels of pSTAT3, as induced by overexpressed leptins, as well as the levels of cyclin D1. In addition to these, IONP-LPrA2 also decreased the progression and proliferation of cells as well as the development of tumorspheres in TNBC cells. Additionally, IONP-LPrA2 exhibited an additive effect on TNBC cells in combination with chemotherapeutics. The chemotherapeutics-loaded IONP–LPrA2 showed a significant reduction of cell survival as compared to chemotherapeutics alone [169]. Liao et al., 2019 fabricated a nanocomposite loaded with nanodiamond-conjugated paclitaxel (ND-PTX) which was further actively targeted by an EGFR inhibitor, cetuximab (Cet), for the treatment of TNBC with overexpressed EGFR. It was observed that the prepared PTX-loaded nanodiamond with functionalized Cet induced a mitotic catastrophe which led to the increased inhibition of cell viability in MDA-MB-231 TNBC cell lines, in comparison to the effects of free PTX, and unloaded nanodiamond. It was further observed that the ND-PTX-Cet distinctively binds with the EGFR and results in apoptosis in EGFR-overexpressed TNBC, as compared to the results in EGFR-negative TNBC. Additionally, ND-PTX-Cet enhanced the expression of caspase-3 and phospho-histone H3 (Ser10) along with causing a significant reduction in tumor volume, as shown in Figure 3 [170]. Wu et al., 2022 engineered ferritin nanoparticles loaded with lapatinib and pseudolaric acid B (PAB) for targeting overexpressed EGFR and CD44 receptors in TNBC. It was observed that the developed ferritin nanoparticles inhibited the EGFR signaling pathway which promoted the LC3B-II expression, thereby facilitating EGFR-mediated autophagy in TNBC cells. Additionally, it was observed that the inhibition of the EGFR signaling pathway by the co-loaded ferritin nanoparticles promoted ferroptosis. It was further found that targeting EGFR and CD44 synergistically mediated the erastin-induced tumor inhibition of cancer stem cell clusters in TNBC. It was also observed that the developed ferritin nanoparticles underwent increased cellular uptake, and showed significant inhibition of tumor growth as evidenced by decreased tumor volume, as shown in Figure 4 [171].

#### 4.2.2. Polymeric Nanoparticles (PNPs)

Polymeric nanoparticles (PNPs) are used quite extensively for the treatment and diagnosis of various cancers. PNPs are fabricated from different types of polymers which either encapsulate the administered drugs or form a covalent bond with the drugs [172]. Various studies observed that PNPs enhance therapy by improving the pharmacokinetics of the anticancer drugs and enhance the drug accumulation to the cancer site by mediating an increased EPR [173]. In PNPs, the polymers used can be of natural origin like chitosan, albumin, gelatin, etc., or synthetic in origin like polylactic acid (PLA), polyethylene glycol (PEG), N-(2-hydroxypropyl)-methacrylamide copolymer (HPMA), poly (D, L-lactide-co-glycolic) acid (PLGA), etc. These polymers are biodegradable and biocompatible [174]. It was also observed that polymers like polyvinyl alcohol (PVA), PEG, etc. are extensively employed in PNPs to evade RES and opsonization, thereby increasing the residence time of the PNPs, further enhancing the therapeutic efficacy against TNBC [173].

Blanco et al., 2014 developed a polymeric micelle loaded with paclitaxel, a chemotherapeutic drug, and rapamycin, an inhibitor of mTOR, for targeting the overexpressed PI3K/Akt/mTOR signaling pathway in TNBC, as shown in Figure 5. From the study, it was observed that the developed nanoparticles showed a significant synergism in vitro, with an evident drug-loading efficacy. It was also observed that after the administration of nanoparticles to mice, the precise drug ratios remained maintained in the tumor for 48 h, offering increased anticancer activity in comparison to what occurred in the control. Mechanistically, it was observed that the simultaneous delivery of drugs suppressed the feedback loop of Akt phosphorylation as well as the associated downstream targets [175].

Similarly, Li et al., 2014 fabricated polymeric micelles loaded with doxorubicin (DOX), which is an anthracycline-based chemotherapeutic, and Decitabine (DAC), which is an inhibitor of DNA methyltransferase (DNMT), for the treatment of TNBC with aberrant DNA hypermethylation. The in vitro studies demonstrated that the nanoparticles loaded with reduced DAC doses and DOX significantly reduced CSCs (cancer stem cells) with increased aldehyde dehydrogenase activity. It was further observed that the DAC-DOX-loaded nanoparticles downregulated DNMT1 and DNMT3b expression and enhanced the expression of caspase-9, which plays an important role in increasing the sensitivity of CSCs towards the treatment. Additionally, the administration of DOX-DAC-loaded nanoparticles induced apoptosis, as shown by decreased tumor volume and average tumor weight [176]. Malarvizhi et al., 2014 developed albumin nano-shell nanoparticles, encapsulating Dasatinib, an inhibitor of Src kinase, which belongs to the family of tyrosine kinase, and a photosensitizer, named m– tetra(hydroxyphenyl)chlorin (mTHPC), for inhibiting the migration of cancer cells in TNBC. It was found from various studies that Dasatinib can prevent the migration of cancer cells. It was observed that the administration of nanoparticles causes the disruption of Src kinase, which impairs cancer cell migration, and that the release of mTHPC produces photoactivated oxidative stress. Such a distinct combination of photo-chemotherapy leads to 99% synergistic cytotoxicity in metastatic TNBC [172]. Bakrania et al., 2018 developed DEAE–Dextran-coated paclitaxel loaded nanoparticles to effectively treat TNBC. It was observed that DEAE–Dextran can induce β-interferon, which significantly decreases ROS generation. Further, the formulations showed increased cellular internalization compared to paclitaxel alone. The study revealed that the combination of paclitaxel and DEAN–Dextran showed synergistic activity through the inhibition of the depolymerization of tubule and the inhibition of the VEGF and NOTCH1 signaling pathways, respectively, as shown in Figure 6 [177].

Sabra et al., 2019 developed magnetic polymeric micelles loaded with Dasatinib (DAS) for the treatment of TNBC. In this study, the magnetic polymeric micelles were prepared using oleic acid-coated magnetite (Fe_3_O_4_) as core, which was further surrounded by amphiphilic zein–lactoferrin. It was observed that the prepared magnetic polymeric micelles demonstrated 10.01 emu. g^−1^ magnetization with a super-paramagnetic property. In addition to these, the formulation also showed improved serum stability and good hemocompatibility in vitro in the presence of sustained DAS release in the acidic pH of the tumor microenvironment. The study further demonstrated that the DAS-loaded magnetic polymeric micelles showed 1.35-fold increased cytotoxicity against MDA-MB-231 cell lines, as compared to the micelles with free DAS, with increased cellular internalization in the presence of an external magnetic field. It was further observed that the magnetic core and lactoferrin corona helped increase the targetability towards the TNBC cells, aided in preventing the cellular migration, and inhibited the expression of p–c–Src protein [178]. Akbarian et al., 2020 engineered human serum albumin nanoparticles (HSA NPs), loaded with artemether (ARM) to improve their bioavailability and therapeutic activity against TNBC. It was observed that the developed HSA NPs increased the water solubility of ARM 50-fold, thereby enhancing its bioavailability. Further, folic acid’s engineered ARM-loaded HSA NPs were surface functionalized to obtain a targeted delivery towards the overexpressed folate receptor (FR-α) in TNBC. The fluorescent microscopy study revealed that the folate-functionalized ARM–HSA NPs underwent increased cellular uptake in FR-α overexpressed TNBC cell lines (MDA-MB-231), compared to low FR-α expressing breast cancer cell lines (SK-BR-3). Further, the cytotoxicity assay demonstrated that the targeted ARM–HSA NPs showed less cell viability than non-targeted ARM–HSA NPs and free ARM. Flow cytometry study indicated that the targeted ARM–HSA NPs exhibited more apoptosis than necrosis [179]. Bhattacharya et al., 2020 fabricated thymoquinone (TQ)-loaded hyaluronic acid (HA)-conjugated polymeric nanoparticles (PNPs) for the effective treatment of TNBC as shown in Figure 7. The formulations were found to be stable at room temperature for approximately 4 months. It was observed that the HA-TQ-NPs showed increased cytotoxicity toward TNBC cells, without showing any harmful effects on healthy cells. Additionally, the HA-TQ-NPs exhibited apoptotic, anti-angiogenic, and anti-migratory activity. It was found that the anti-migratory effect of the developed HA-TQ-NPs in the TNBC was due to the upregulation of microRNA-361, which eventually downregulated the migratory factors, namely Rac1 and Rho A. It was further observed that the developed formulation decreased the secretion of VEGF-A, thereby preventing the development of tumor-induced vascularization [180].

Qin et al., 2020 developed polymeric micelles loaded with a combination of paclitaxel and sunitinib for enhanced synergistic anticancer activity against TNBC. It was demonstrated that sunitinib functions as an inhibitor of tyrosine kinase receptors. Hence, it aids in targeting the kinase signaling pathways responsible for the progression of TNBC. It was observed that after loading a 1:5 ratio of paclitaxel and sunitinib into the polymeric micelles, the micelles showed synergistic anticancer activity, which increased apoptosis. It was further observed that the synergistic anticancer activity of the loaded polymeric micelles was due to the induction of an immunogenic cell death (ICD) response. Additionally, the loaded polymeric micelles exhibited enhanced tumor immunogenicity and promoted various immunosuppressive factors within the tumor microenvironment [181]. Misra et al., 2021 prepared PLGA nanoparticles in association with RNA interference technology (siRNA), encapsulating a chemotherapy drug, paclitaxel, and an inhibitor of PARP activity, olaparib, for effective and synergistic anticancer activity, and increased targeting to the FOXM1 proto-oncogenic transcription factor in TNBC. It was found from the study that the co-delivery of paclitaxel and olaparib, in combination with FOXM1-siRNA by the nanoparticles, increased the anticancer efficacy against TNBC, as observed by cytotoxicity study and apoptosis study, in comparison to the effects of free drugs. In addition, the flow cytometry data demonstrated that the siRNA-NPs showed increased cellular accumulation compared to native siRNA [182]. Khesht et al., 2021 developed chitosan lactate nanoparticles for the delivery of doxorubicin and CD73 siRNA to the TNBC cells. The developed nanoparticles were further surface functionalized by TAT (transactivating transcriptional activator) peptide derived from HIV-1 and hyaluronate (HA) for enhanced targetability. It was observed that on combination delivery, the dose of doxorubicin was reduced, which in turn decreased the side effects of chemotherapeutic drugs. Additionally, the combinatorial delivery of doxorubicin and CD73 siRNA inhibited cell proliferation and migration as shown in Figure 8. Further, the developed nanoparticles decreased tumor volume, enhanced the survival rate for tumor-bearing mice, and induced anti-tumor immune responses [183].

Chen et al., 2021 fabricated polymeric nanoparticles loaded with CD155 siRNA and surface functionalized with PD-L1 antibodies for targeting the overexpressed PD-L1 and CD155, and providing effective anticancer activity against TNBC. It was observed that the combination delivery asynchronously blocked the expression of PD-L1 and CD155 in a spatiotemporal manner. It was observed that the developed polymeric nanoparticles increased the early-stage CD8+ T cell immune surveillance against TNBC while reversing the inhibition profile of the late-stage CD8+ T cells in TNBC cells to prevent the tumor immune escape. Additionally, the combinatorial delivery induced immunogenic cell death (ICD) within the TNBC cells to boost immune checkpoint therapy. Additionally, the developed polymeric nanoparticle inhibited the proliferation and migration of TNBC cells [184]. Zeng et al., 2022 engineered polymeric nano micelles loaded with olaparib and hyaluronic acid-conjugated dasatinib for increasing their therapeutic activity against TNBC. It was observed that the loading of the two drugs within the nanoformulation increased their aqueous solubility, improving their bioavailability in the TNBC cells. Additionally, it was observed that the loaded nano micelles exhibited increased cellular uptake due to their binding with the overexpressed CD44 proteins on the surface of the TNBC cells. The nanoformulations further showed apoptosis by inducing DNA damage, preventing DNA damage repair, and inhibiting PARP expression. The nano micelles also prolonged the circulation time of the drugs in the blood, thereby improving their bioavailability and increasing their therapeutic efficacy for the treatment of TNBC [185].

#### 4.2.3. Lipid-Based Nanoparticles (LNPs)

Apart from PNPs, lipid-based nanoparticles (LNPs) are also at the frontline of the delivery system and are used to significantly enhance the pharmacokinetic and bioavailability attributes of drugs with a simultaneous decrease of the side effects. LNPs like liposomes, emulsions, nanocapsules, nanospheres, solid lipid nanoparticles, nanostructured lipid carriers, and exosomes are employed in drug delivery and the diagnosis of cancer [186,187]. LNPs are spherical nanovesicles comprised of phospholipids, cholesterol, and solid and liquid lipids, and are used to entrap hydrophobic drugs as well as hydrophilic drugs. In addition to these, they also contain surfactants and co-surfactants used to reduce the interfacial tension between two phases and form a monophasic layer. Various anticancer therapeutics have already been formulated as liposomal formulations such as lipoplatin, which is a cisplatin-loaded liposome [188]. Various studies inferred that compared to PNPs, LNPs are considered more beneficial for treatment, because the LNPs are composed of GRAS-recognized biocompatible and biodegradable lipids, making them safe for delivery [189].

Guo et al., 2016 developed a liposomal formulation, encapsulating Lipocalin 2 (Lcn2) siRNA for the treatment of TNBC with an overexpressed ICAM-1 molecular target. It was observed that the ICAM-1-targeted Lcn2-liposomes mediated a strong interaction or binding with the TNBC cells (MDA-MB-231) compared to the interaction of the non-cancerous cells (MCF–10A) with the TNBC cells. Additionally, it was observed that the knockdown of Lcn2 through ICAM-1-targeted Lcn2-liposomes resulted in a significant decrease in the generation of VEGF (Figure 9), which led to decreased angiogenesis both in vitro as well as in vivo [190].

Zhang et al., 2019 prepared nanoparticles loaded with paclitaxel amino lipid derivative and P53 mRNA for the effective treatment of TNBC by specifically targeting the mutated TP53. It was observed that the PAL-P53 mRNA NPs exhibited superior characteristics in comparison to Abraxane^®^ and Lipusu^®^. The developed loaded NPs also showed synergistic cytotoxicity in vitro, with an effective anti-tumor efficacy in vivo [191]. Burande et al., 2020 developed Cetuximab-functionalized liposomes loaded with paclitaxel and piperine for the treatment of EGFR-positive TNBC. The study observed that the targeted liposomes showed increased cellular internalization of paclitaxel and piperine and cytotoxicity compared to non-targeted and free drugs. In addition, the combination-loaded formulation showed significant synergistic activity [192]. Li et al., 2020 developed macrophage-derived exosomes which were further coated with poly (lactic-co-glycolic acid) and loaded with doxorubicin. Further, to increase their targetability, the exosomes were surface functionalized with a peptide for targeting c-Met, a factor responsible for mesenchymal-epithelial transition (MET). It was revealed from various studies that c-Met is found to be overexpressed in TNBC. The study observed that the targeted exosomes exhibited increased cellular internalization and anticancer efficacy through increased apoptosis and enhanced the inhibition of tumor growth [193]. Darabi et al., 2022 fabricated doxorubicin-loaded solid lipid nanoparticles with surface functionalization with anti-EGFR/CD44 dual RNA aptamers to effectively treat TNBC. For effective nuclear delivery, the fabricated surface-functionalized SLN was further chemically conjugated with dexamethasone. From the in vitro drug release study, it was observed that 96.1 ± 1.97% doxorubicin was released from the SLN within 48 h of incubation. Additionally, the DEX-DOX-SLN with anti-EGFR/CD44 aptamers exhibited enhanced inhibition of cell proliferation compared to DEX-DOX-SLN without any aptamers to inhibition of migration, and angiogenesis [194]. Zhu et al., 2022 fabricated gemcitabine and paclitaxel-loaded lipid nanoparticles, with surface conjugation via the ICAM-1 binding peptide LFA1–P for increased targeting and effective treatment of TNBC. It was observed that the peptide conjugation showed 4-fold increased binding of the nanoparticles compared to that of the non-conjugated nanoparticles. Additionally, the peptide-conjugated nanoparticles showed increased cellular internalization and 60-times-increased target/healthy tissue (lung/gastrointestinal (GI)) ratio compared to non-conjugated nanoparticles. Further, it was observed that the peptide-conjugated nanoparticles prevented cancer cell metastasis to the lungs [195].

Currently, scientists are focusing more on green chemistry and green science, where more sustainable and natural compounds are synthesized, thereby mimicking biological or living conditions to a maximum extent. In recent times, natural materials are combined with synthetic NPs to bio-mimic nanoparticles into biological systems, leading to increased drug targetability and reduced immunogenicity. In this context, researchers are extracting natural cell membranes and coating synthetic NPs with them, thus facilitating enhanced biofunctionalization. It was observed that cell membrane coating preserved the physiochemical integrity of the NPs while exerting intrinsic and complex cellular functions. Additionally, it was observed that for mediating enhanced targeting, the NPs were coated with the membrane derived from specific cells, and such targeting was observed through homotypic and heterotypic adhesion [196]. Further, it was summarized that the cell membrane-coated NPs facilitated advanced immune responses, bypassed clearance via RES, prolonged circulation, and enhanced targetability [197]. On this note, Jin et al., 2021 engineered cancer cell membrane (CM)-coated nanoparticles loaded with doxorubicin which was conjugated with reactive oxygen species (ROS)-sensitive polymer. The engineered nanoparticles were further surface functionalized by the anti-CD73 antibody and were further combined with photodynamic therapy to enhance their anti-metastatic activity and ensure increased anticancer activity. It was observed that due to the presence of CM, the nanoparticles underwent an immune escape from the macrophages. The existence of anti-CD73 provided an immunosuppressive activity by blocking the adenosine pathway. It was further observed that the combination of chemotherapy and photodynamic therapy also prevented abscopal tumor metastasis and inhibited orthotopic tumors. Additionally, the combination of chemotherapy and photodynamic therapy demonstrated a 72.5% tumor inhibition rate, as compared to chemotherapy with laser treatment, which demonstrated 58.7%, and when the combination of chemotherapy and photodynamic therapy was applied in combination with anti-CD73 antibody, the tumor inhibition rate was increased to 93.4%, which inferred that the therapy not only destroyed the cancer cells by generating ROS but also reversed the immunosuppression to elevate the anti-tumor immunity [198].

Table 3 summarizes the different biomarker-based nanoparticles developed for the treatment and diagnosis of TNBC.

Although NPs have emerged as an efficient drug delivery system over conventional delivery systems due to their characteristics (mentioned above) and advantages, a certain fraction of them reaches the clinics. It was further found that due to the presence of a mononuclear phagocytic system, most of the NPs get accumulated in off-target organs after administration, rather than in the target site. One of the reasons for such a scenario is the prevalence of “protein corona (PC),” which is described as the array of proteins attached to NPs, affecting its colloidal stability, optimal biodistribution, interactions, and clearance [199]. Moreover, it was established that the assembly of PC and NPs was the first interaction that the NPs experience after entering the body. Thus, to overcome such interactions, various studies were performed, wherein PEGylation was found to be the widely used strategy. But PEGylation also shows some hindrances including a decrease of cellular uptake of NPs by the target cells and the formation of circulating anti-PEG antibodies on the excessive administration of PEGs, thus creating a state of hypersensitivity and compromising the safety of the NPs [200]. Henceforth, such synthetic strategies were overtaken by the biomimetic approaches which include a coating of NPs with cell membrane, application of the virus or its components, and manipulating the integrity of PC [201].

It was further revealed that coating the NPs with cell membrane makes the NPs inherent biological identities, aiding the NPs in avoiding opsonization as well as the production of anti-NP immunoglobulins and thus, further facilitating the rapid clearance of NPs [196]. It was not unnoticed that viruses could be considered in the design of drug delivery systems. Viruses and their components were found to interact with the proteins in biological fluids. Berardi, et al., 2019 found that cowpea mosaic virus (CPMV)-coated NPs and bluetongue virus (BTV)-coated NPs exhibited the existence of less PC compared to standard polystyrene NPs [199]. Additionally, it was found that apart from a coating of NPs with cell membrane and viruses, the manipulation of PC attached to NPs also plays an important role in diminishing its interaction with the NPs. The PC environment can be manipulated by attaching various stealth-inducing biological materials like alginic acid or hyaluronic acid to the NPs. Such attachments alter the sizes, surface potentials, and stabilities of the NPs [202].

Hence, it could be inferred from the above paragraph that the interaction of PC and NPs can be manipulated by understanding the PC’s composition and integrity developed around the NPs. From this perspective, it becomes essential to develop a specific harmonized technique for analyzing the assembly of PC.

## 5. Clinical Status of TNBC Biomarkers-Based Nanotherapeutics

Nanotechnology has driven forward the application of nanoparticles for the diagnosis and treatment of TNBC, where lipid-based nanoparticles, usually liposomes, and polymeric nanoparticles constitute the main types of nanoparticles that have been translated from the in vitro studies to the clinical studies [164]. A new wave of nanoformulation development is anticipated to incorporate the molecular biomarkers of individuals into nanoparticles to offer a precise, personalized, and selective delivery to TNBC cells. Such nanoformulations unlike conventional therapies can be developed as more targeted therapies. However, despite all the merits of nanoformulations over conventional therapies, the number of nanoformulations reaching the clinics is still very low compared to the number of nanoformulations developed at the pre-clinical level, which has been found to be much smaller for those nanoformulations that finally get approval. Such a crisis has resulted due to a lack of clinical efficacy, toxicity, and a complex manufacturing setup. In this context, major efforts are being applied to establish pre-clinical models that can mimic the clinical criteria much better, facilitating more “realistic” results in cases of efficacy and toxicity. In the same pipeline, various new techniques and technologies are being developed to simplify the large-scale manufacturing of nanoformulations that will permit the fabrication of reproducible formulations. It could be assumed that the mentioned challenges could be overcome in the following decades, at least in part. Table 4 lists the clinical trials that have studied the effects of biomarker-based nanoparticles against TNBC [203].

## 6. Conclusions and Future Perspective

From the present review, we concluded that extensive advancement has been made to understand the biology of cancer, which further led to a recognition of the heterogenous nature of TNBC. Such heterogenous characteristics arose due to the presence of six subtypes of TNBC, further leading to the identification of various prognostic and predictive biomarkers as evident by the activation or inactivation of distinct signaling pathways, receptors, genes, etc. As specific biomarkers characterize each subtype, targeting those biomarkers results in precise and personalized TNBC therapy, thereby avoiding intraspecies variability. Further, biomarker screening has been proven an important strategy for characterizing distinct cancer types and cellular aberrations that trigger cancer development. However, despite such advancements, it has been demonstrated that chemotherapy remains the backbone of TNBC therapy, as TNBC cells are insensitive to endocrine treatment. Thus, to broaden the treatment regimen of TNBC, and escape the off-target side effects of chemotherapy, various targeted therapies have been adopted based on the biomarkers. In the present article, we have discussed various signaling pathways, aberrant gene expressions, receptors responsible for TNBC progression, and the therapeutics assigned for their treatment.

It was further observed that the conventional drug delivery system has proven inefficient in delivering targeted therapy; moreover, it has its own drawbacks, like low efficacy, non-tunability, etc. All these characteristics have been observed in the case of nanoparticles which offer surface modification or functionalization, increased targeting, and enhanced efficacy. The application of nanotechnology in the precise delivery of anticancer drugs, molecules like DNA, siRNA, etc. further enhances the chances of therapeutic success and helps in integrating the molecular biomarkers for the distinctive recognition of biomolecules that are specific to each subtype of TNBC. It was thus concluded that conjugating these biomarkers to nanotechnology-based platforms critically improves anticancer therapy efficiency and TNBC management.

Despite the advantages of nanoparticles in treating and diagnosing TNBC, a very limited number of nanotherapeutics find their way to the clinics, mostly due to their complex design, a lack of expertise in the manufacturing sector, and regulating guidelines, costs, and testing parameters. It was further suggested that as nanotherapeutics are emerging as a pivotal platform for diagnosis and treatment, updated evaluation and policy regarding nanotherapeutics are being established, clearing the roadblocks responsible for not letting nanotherapeutics into clinics. Additionally, it was observed that as artificial intelligence (AI) is getting associated with every field of life, it was assumed that soon, scientists will be incorporating artificial intelligence technologies in nanotherapeutics as well as biomarker screening and targeting. AI will employ extensive algorithms to extract more precise and exhaustive information regarding the biomarkers present in the heterogenous population of TNBC patients. This will save time, manpower, and resources in doing the same. We thus believe that, in the coming future, AI will help in identifying more new biomarkers and developing the latest biomarkers-driven nanotherapeutics. Such innovations have led to the development of bioinformatics. It was observed that with bioinformatics, one can predict the origin of cancer and its mRNA expression, thus paving the way to a whole new world of cancer-genomics, which will later aid in developing new drugs and their delivery systems.

## Figures and Tables

**Figure 1 cancers-15-02661-f001:**
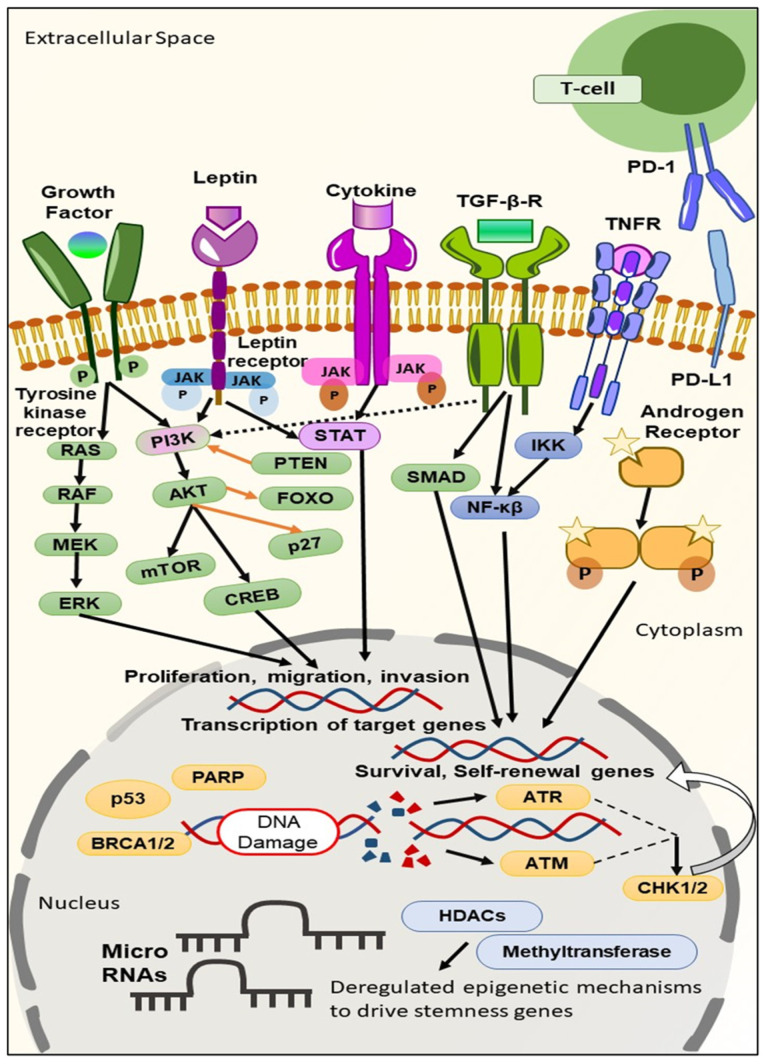
Different signaling pathways and epigenetic mechanisms that have been deregulated during TNBC progression and growth, contributing to stemness. Tyrosine kinase receptors promote tumorigenesis through Ras and the PI3K/AKT/mTOR signaling pathway. ERK phosphorylation, the activation of STAT, and the PI3K/AKT/mTOR pathway promote EMT, and regulate the proliferation, migration, and invasion of cancer cells. The activation of NF-κβ and the SMAD pathway promote the survival and self-renewal of genes. In the cytoplasm, the androgen receptor (AR) binds with the chaperone proteins and undergoes phosphorylation, which promotes the transcription of target genes in the nucleus. Within the nucleus, at the genetic level, the BRCA1/2, and p53 mutation, promote TNBC progression. PD-1/PD-L1 signaling suppresses CD8+ T activation, which in turn results in a tumor microenvironment and decreases tumor-infiltrating lymphocytes. The inactivation of PTEN, and FOXO, promote the PI3K/AKT/mTOR signaling pathway.

**Figure 2 cancers-15-02661-f002:**
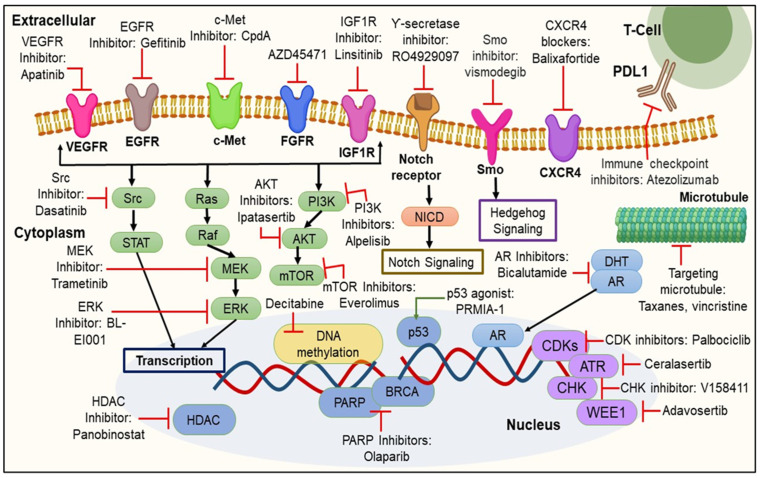
Potential therapeutic targets and their associated drugs for the treatment of TNBC. The black arrow shows upregulation or excitatory regulation, and the red-headed line bar shows the corresponding inhibitory effect.

**Figure 3 cancers-15-02661-f003:**
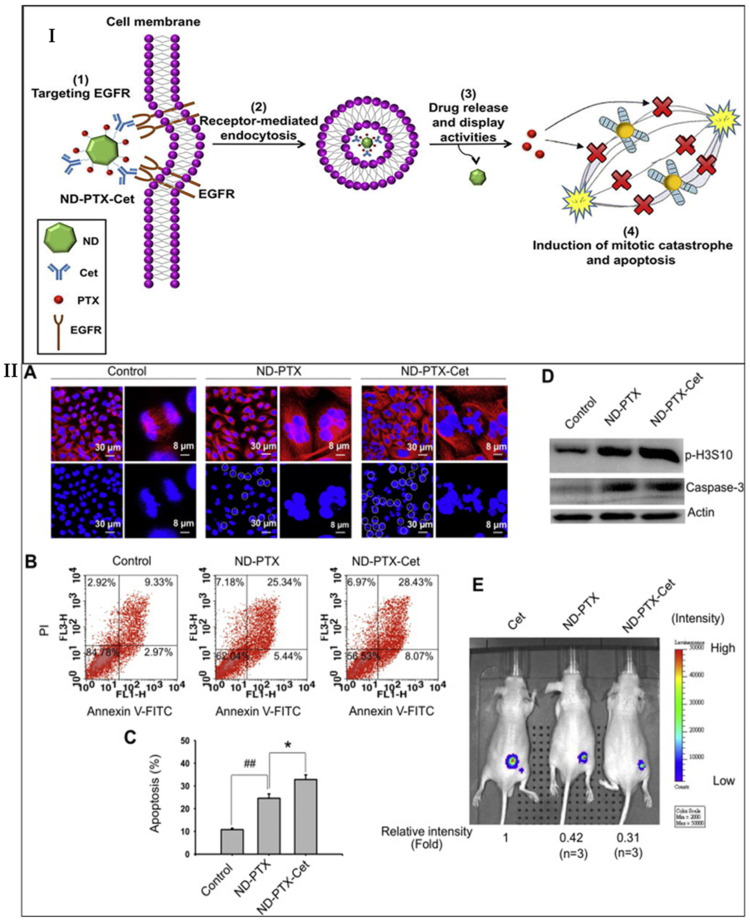
(**I**) The model of ND-Cet-PTX, used in enhancing drug efficacy and overcoming drug resistance in human triple-negative breast cancer. (**II**) ND-PTX-Cet enhanced the mitotic catastrophe, apoptosis, and tumor inhibition. (**A**) The MDA-MB-231 cells were treated with or without 1 mg/mL of ND-PTX and ND-PTX-Cet for 48 h. (**B**) Flow cytometry analysis: Cells staining with Annexin V+/PI are those undergoing early apoptosis (lower right), and Annexin V+/PI+-stained cells are undergoing late apoptosis (upper right). (**C**) The population of total apoptotic cells (including early and late apoptotic cells) was quantified using CellQuest software. The bar represents the mean ± S.E. ## *p* < 0.05, indicating a significant difference between the ND-PTX and control samples. * *p* < 0.05 indicates a significant difference between the ND-PTX and ND-PTX-Cet treated samples. (**D**) The protein levels of p-Histone H3 (Ser10) and active caspase-3 were determined by the western blot. Representative images of the western blot are shown from one of three independent experiments. (**E**) The nude mice were subcutaneously injected with 4 × 10^6^ MDA-MB-231-luc2 tdTomato breast cancer cells. After inoculation, the mice bearing tumors were treated with or without 20 mg/kg of Cet, ND-PTX, or ND-PTX-Cet three times. The luminescence intensities of MDA-MB-231-luc2 tdTomato tumors were observed under the IVIS system at 16 days. The luminescence intensity of tumors was quantified by the IVIS system using the analysis by the Xenogen Living Image software, Version 4.0. The results were obtained from three groups. “Reprinted/adapted with permission from Ref. [170]. 2019, Wei-Siang Liao”.

**Figure 4 cancers-15-02661-f004:**
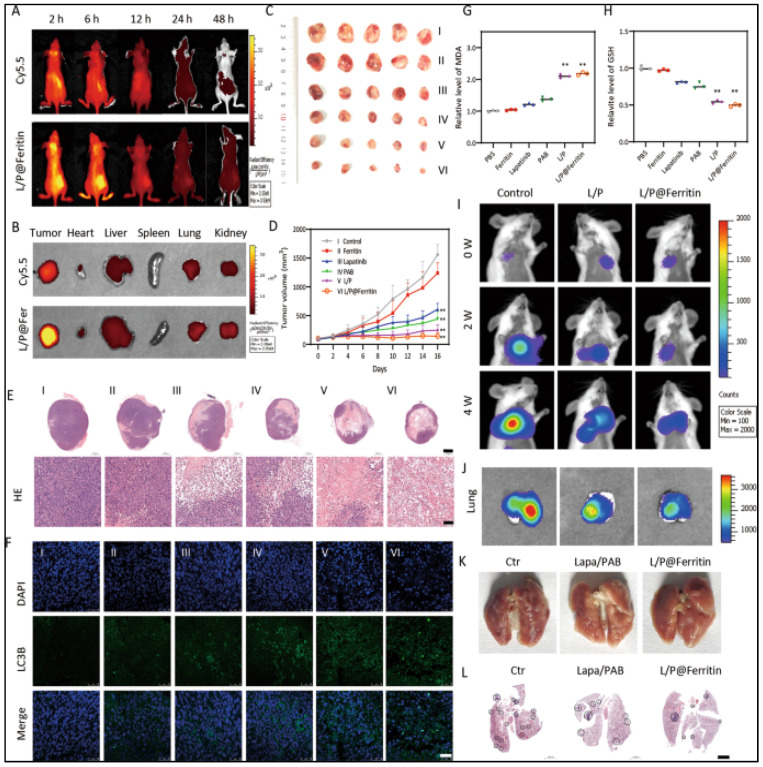
L/P@Ferritin inhibited tumor growth in vivo. (**A**,**B**) Fluorescence of L/P@Cy5.5-Ferritin in tumor-bearing mice at different times, and in tumor and organs which were harvested at 6 h. (**C**,**D**) L/P@Ferritin effectively inhibited xenograft tumor growth. I, II, III, IV, V, and VI represent the groups of PBS, ferritin, lapatinib, PAB, L/P, and L/P@Ferritin, respectively. (**E**) HE stain of xenograft tumor; I, II, III, IV, V, and VI represent the groups of PBS, ferritin, lapatinib, PAB, L/P, and L/P@Ferritin, respectively; scale bars: up panel, 2 mm; low panel, 100 μm. (**F**) Expression of LC3 detected by IF; scale bars represent 50 μm. (**G**,**H**) Overgeneration of MDA and depletion of GSH in xenograft tumor treated with L/P@Ferritin. (**I**,**J**) Bioluminescence images of MDAMB-231-Luc cells in mice and extracted lungs in control, L/P, and L/P@Ferritin groups, respectively. (**K**) Numbers of pulmonary metastatic nodules in control, L/P, and L/P@Ferritin groups, respectively. (**L**) HE stain of the lung in control, L/P, and L/P@Ferritin groups, respectively; scale bars represent 2 mm. ** *p* < 0.01. “Reprinted/adapted with permission from Ref. [171]. 2022, Xinghan Wu.”

**Figure 5 cancers-15-02661-f005:**
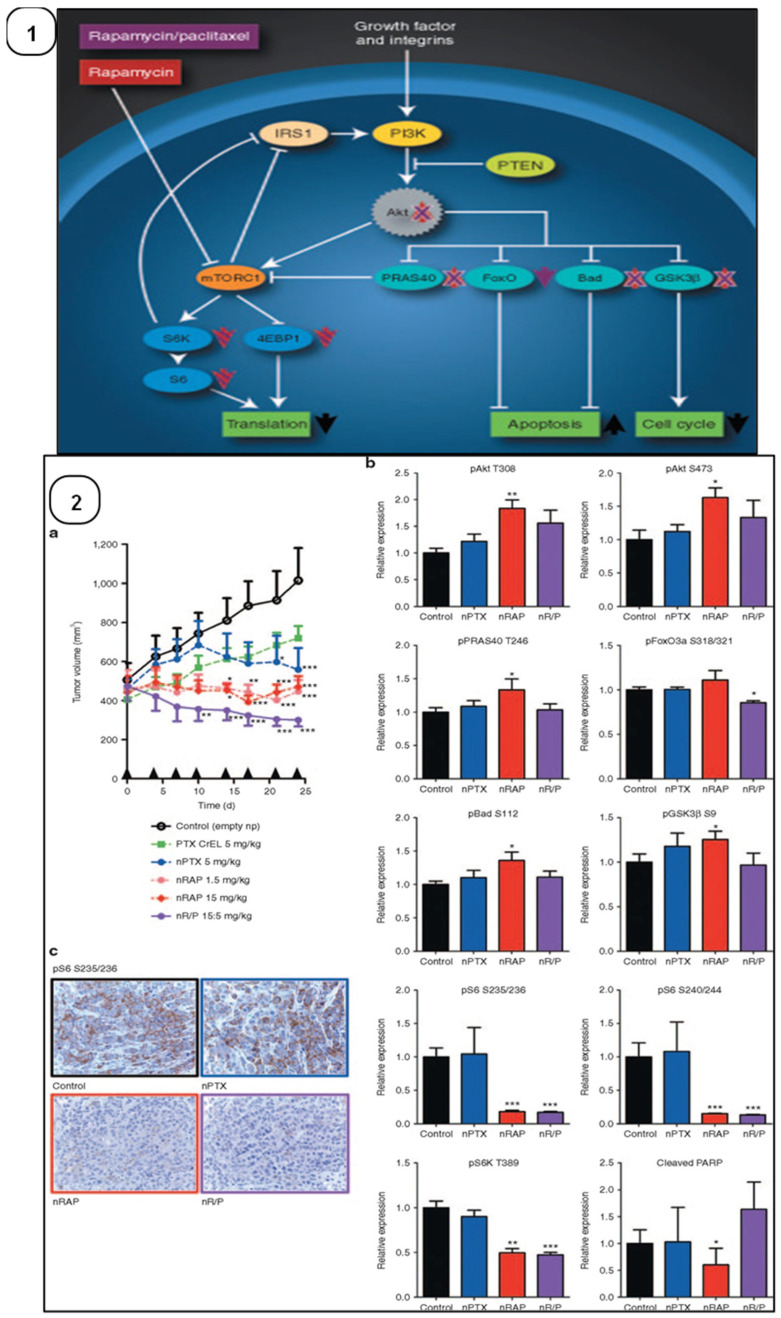
(**1**) Illustration indicating the mechanism of Rapamycin and paclitaxel nanoparticles in synergistically targeting the PI3K/Akt/mTOR pathway via the suppression of Akt phosphorylation. (**2**) In vivo anticancer activity of Rapamycin and paclitaxel nanoparticles by targeting the PI3K/Akt/mTOR signaling pathway: (**a**) Inhibition of tumor growth after the administration of rapamycin nanoparticles (nRAP), paclitaxel nanoparticles (nPTX) and rapamycin-paclitaxel combination nanoparticles (nR/P) to MDA-MB-468 tumors-induced mice (mean ± SEM, n = 5). * Denotes that the results obtained are statistically significant as compared to what was found in the control group (* *p* < 0.05; ** *p* < 0.01; ***, *p* < 0.001). (**b**) Relative expression of proteins involved in the PI3K/Akt/mTOR pathway as determined by the reverse-phase protein array analysis of tumors excised 24 h after treatment (mean ± SEM; n = 5). * Indicated statistical significance in comparison to control (* q < 0.1; ** q < 0.05; *** q < 0.005). (**c**) pS6 S235/236 immunohistochemical staining of excised tumors 24 h after administration of nanoparticles. The scale bar represents 20 µm. “Reprinted/adapted with permission from Ref. [175]. 2014, Elvin Blanco.”

**Figure 6 cancers-15-02661-f006:**
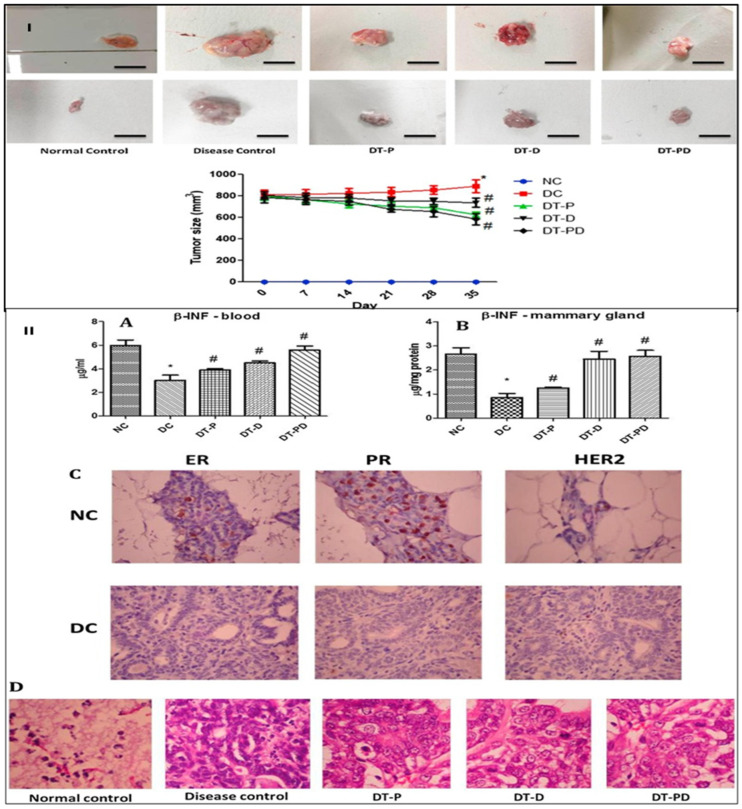
(**I**) Tumor regression analysis in xenograft model over treatment duration. * Significantly different from normal control (*p* < 0.05), # Significantly different from disease control (*p* < 0.05). Scale bar represents 10 mm. (**II**) (**A**) Determination of β-interferon release in various treatment groups in a xenograft model in blood and (**B**) mammary glands; (**C**) Immunohistochemistry studies for ER, PR, and HER2 in mammary gland sections; (**D**) Histopathological studies of nanoformulation-treated mammary gland in the xenograft model. * Significantly different from normal control (*p* < 0.05), # Significantly different from disease control (*p* < 0.05). Magnification ×100. NC: normal control, DC: disease control, DT-P: disease treated with paclitaxel nanoparticles, DT-D: disease treated with DEAE–Dextran, DT-PD: disease treated with DEAE–Dextran-coated paclitaxel nanoparticles, ER: estrogen, PR: progesterone and HER2 receptors. “Reprinted/adapted with permission from Ref. [177]. 2018, Anita K. Bakrania.”

**Figure 7 cancers-15-02661-f007:**
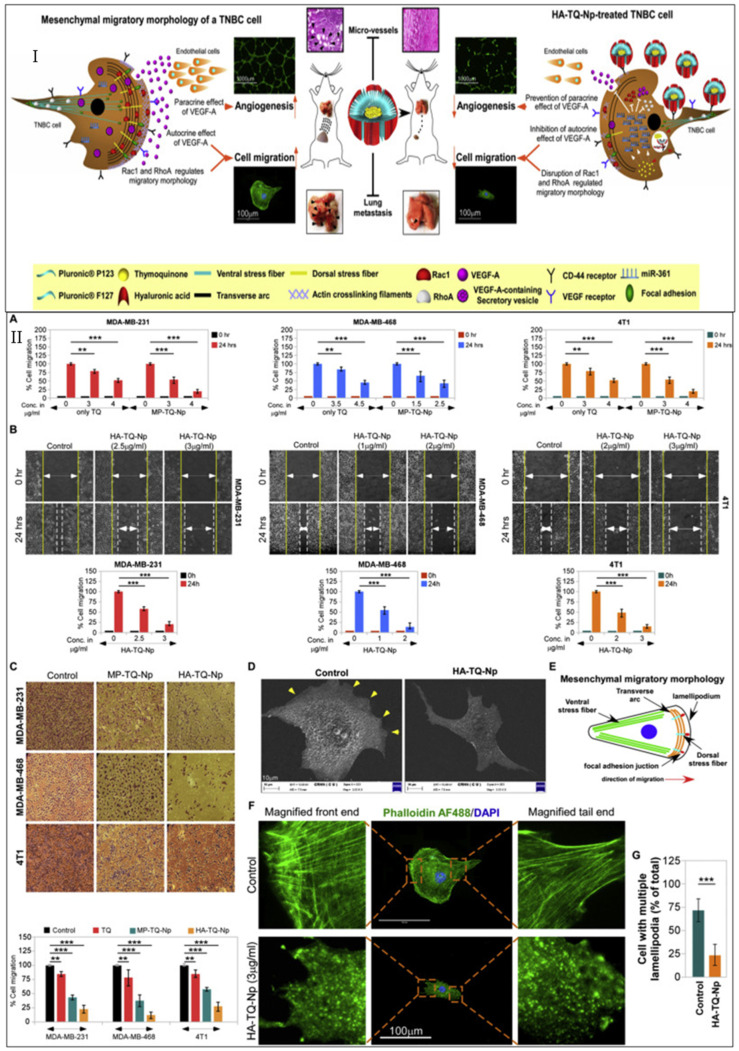
(**I**) Schematic illustration depicting the molecular mechanisms of HA-TQ-Np mediated anticancer effect on TNBC cells. (**II**) Elucidation of the anti-migratory effect of HA-TQ-Nps on TNBC cell lines. (**A**) The graphical representations of the percent cell migrations of MDA-MB-231 (left panel), MDA-MB-468 (middle panel), and 4T1 (right panel) cells upon treatment with different doses of free TQ and MP-TQ-Nps at 0 and 24 h of incubation. (**B**) The pictorial (upper panels) along with graphical (lower panel) representations of the bidirectional wound-healing assay, illustrating the rate of migration of MDA-MB-231 (left panel), MDA-MB-468 (middle panel) and 4T1 (right panel) cells upon treatment with different doses of HA-TQ-Nps at 0 and 24 h of incubation. Magnification: 20×. (**C**) The phase-contrast images (upper panel) and graphical depictions of percentage cell migration (lower panel) evaluated through transwell migration assay for the untreated (control), free TQ-, MP-TQ-Np- and HA-TQ-Np-treated three aforementioned TNBC cell lines at their respective migratory doses for each treatment modality for 24 h. Magnification: 20×. (**D**) The micrographs of the cellular morphology of untreated and HA-TQ-Np-treated MDA-MB-231 cells were visualized under SEM. The yellow arrows on the control cells point at the multiple lamellipodia seen on the cellular surface. The scale bar is 10 μm. (**E**) The schematic drawing of the mesenchymal morphology of a migratory cell where the organization of all the actin stress fibers is pointed out. (**F**) The fluorescent micrographs of both the untreated and HA-TQ-Np-treated MDA-MB-231 cells. The green fluorescence of Alexa Flour 488 (AF488)-conjugated Phalloidin represents actin filaments. The blue fluorescence of DAPI represents viable nuclei. Magnification: 40× and the scale bar is 100 μm. (**G**) The graphical illustration of the differential percentage of MDA-MB-231 cells with multiple lamellipodia in both the untreated and HA-TQ-Np-treated conditions. Each value is depicted as Mean ± SD; n = 3. ** *p* < 0.01, and *** *p* < 0.001. “Reprinted/adapted with permission from Ref. [180]. 2020, Saurav Bhattacharya”.

**Figure 8 cancers-15-02661-f008:**
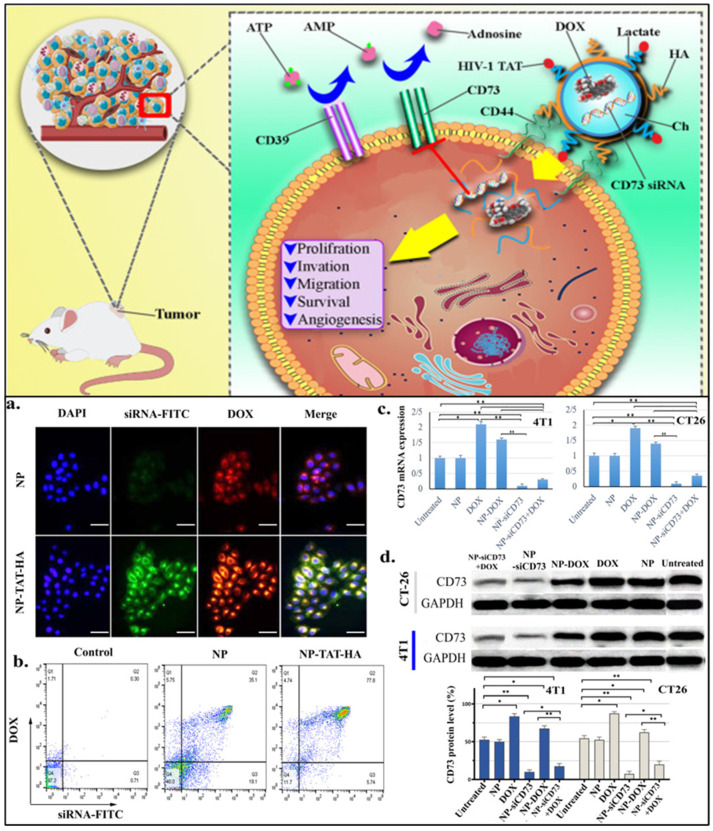
A blockade of CD73 using siRNA-loaded CL NPs functionalized with TAT-HA and loaded with DOX can effectively prevent tumor growth. CL-TAT-HA NPs can deliver DOX and anti-CD73 siRNA to cancer cells and significantly suppress the survival, invasion, proliferation, and migration of cancer cells (Top). The cellular uptakes of TAT-HA-conjugated NPs and non-targeted NPs were examined by confocal microscopy (**a**) and flow cytometry (**b**). The impact of siCD73 and DOX-loaded CL-TAT-HA NPs on the CD73 expression in cancer cells was investigated by using qPCR (**c**) and the western blot (**d**) (Bottom). * *p* < 0.1, and ** *p* < 0.01. “Reprinted/adapted with permission from Ref. [183]. 2021, Armin Mahmoud Salehi Khesht.”

**Figure 9 cancers-15-02661-f009:**
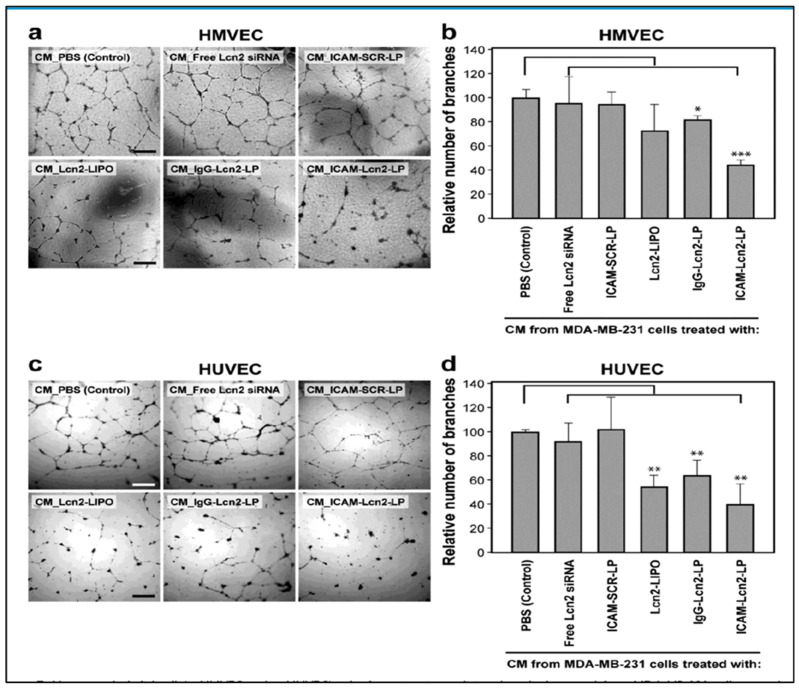
Human endothelial cell ((**a**) HMVEC and (**c**) HUVEC) tube formation in conditioned media harvested from MDA-MB-231 cells treated with immunoliposomes. The number of branches was quantified for (**b**) HMVEC and (**d**) HUVEC. All scale bars are 200 µm (* *p* < 0.05, ** *p* < 0.01, *** *p* < 0.001). “Reprinted/adapted with permission from Ref. [190]. 2016, Peng Guo.”

**Table 1 cancers-15-02661-t001:** Clinical trials of biomarkers in TNBC.

Agents	Biomarkers (Targeting Moiety)	Clinical Phase	Identifier
APR-246 + Pembrolizumab	TP53 + PD-1	I/II	NCT04383938
Ribociclib + Bicalutamide	CDK4/6	I/II	NCT03090165
Taselisib + Enzalutamide	PI3K/AKT/mTOR	I/II	NCT02457910
Alpelisib + Enzalutamide	PI3K/AKT/mTOR	I	NCT03207529
Olaparib + Carboplatin/Paclitaxel	PARP	I	NCT00516724
MEDI4736 + Olaparib and/or Cediranib	PD-L1 + PARP + VEGFR	I/II	NCT02484404
Olaparib + Durvalumab	PARP + PD-L1	II	NCT03801369
Talazoparib	PARP	II	NCT03901469
Olaparib + Onalespib	PARP + HSP90	I	NCT02898207
HX008 + Niraparib	PD-1 + PARP	II	NCT04508803
Prexasertib	CHK1	II	NCT02873975
IDX-1197	PARP	I/II	NCT04174716
Avelumab	PD-L1	II	NCT02554812
Nivolumab + Bicalutamide + Ipilimumab	PD-1 + AR + CTLA4	II	NCT03650894
Avelumab + Binimetinib, Utomilumab, or anti-OX40 antibody	PD-L1 + MEK ½, CD 137 or OX40	II	NCT03971409
Atezolizumab in different combinations	PD-L1 in different combinations, including chemotherapy, ADC, CD40, IL6R, VEGFA, and AKT	I/II	NCT03424005
Spartalizumab + LAG525 in combination with NIR178, Capmatinib, MCS110, or Canakinumab	PD-1 + LAG-3 in combination with anti- adenosine A2A receptor, Met receptor, CSF-1 or IL1β	I	NCT03742349
Sacituzumab govitecan + Talazoparib	ADC + PARP	I/II	NCT04039230
AMXI-5001	PARP and a microtubule polymerization inhibitor	I/II	NCT04503265
BKM120/BYL719 + Olaparib	PI3K + PARP	I	NCT01623349

**Table 3 cancers-15-02661-t003:** Biomarker-based nanoparticles for treatment and diagnosis of TNBC.

S. No.	Formulation	Targeting Biomarkers	Agent/Drug	Results	Ref.
1	Polymeric micelles	mTOR inhibitor: Rapamycin	Paclitaxel	- Synergistic anticancer activity was observed - Simultaneous delivery suppressed the phosphorylation of the Akt loop and its downstream targets	[175]
2	Polymeric nanoparticles	DNA hypermethylation inhibitor: decitabine	Doxorubicin	- Nanoparticles exhibited a reduced proportion of CSCs with increased activity of aldehyde dehydrogenase - Co-delivery downregulated the expression of DNMT1 and DNMT3b and upregulated the expression of caspase-9	[176]
3	PLGA polymeric nanoparticles	tyrosine kinase inhibitor, Dasatinib	Photo-sensitizer: m-tetra (hydroxyphenyl) chlorin (mTHPC),	- The nanoparticles generated photoactivated oxidative stress which disrupted the Src kinase, thereby preventing the migration of cancer cells - Additionally, co-delivery resulted in 99% (approximately) synergistic cytotoxicity	[172]
4	pH-responsive liposomes	ICAM-1 antibody	Lipocalin 2 (Lcn2) siRNA	- Targeted liposomes reduced the formation of VEGF, which resulted in a reduction of angiogenesis	[190]
5	Multi-branched gold nanoantennas (MGN)	anti-PDL1 antibodies, and anti-EGFR antibodies	Dithio-bis-(2-nitrobenzoic acid), and pMBA (para mercaptobenzoic acid)	- MGN underwent increased localization via surface binding as well as receptor-mediated endocytosis	[168]
6	Iron oxide nanoparticles	Leptin antagonist: LPrA2	Cisplatin, Doxorubicin, Cyclo-phosphamide, and Paclitaxel	- Nanoparticles reduced the levels of leptin-activated pSTAT3 and cyclin D1 - Co-delivery decreased the survival of cancer cells	[169]
7	Polymeric nanoparticles	VEGFR inhibitor: DEAE-Dextran	Paclitaxel	- Nanoparticles exhibited synergistic anticancer activity - Targeted nanoparticles also inhibited ROS generation by inducing β-interferon	[177]
8	Oleic acid-coated Magnetite (Fe_3_O_4_) based polymeric micelles	Lactoferrin	Dasatinib	- The developed nanoparticles showed a 1.35-fold-increased cytotoxicity - The nanoparticles also prevented the expression of p-c-Src protein induced the inhibition of cellular migration, as well as increasing the targetability due to the presence of lactoferrin corona	[178]
9	Nano-composite	EGFR inhibitor: cetuximab	Paclitaxel	- The nanoparticles showed decreased cell viability with increased cellular internalization, apoptosis, and mitotic catastrophe	[170]
10	Lipidic nanoparticles	P53 mRNA	Paclitaxel amino lipid	- The nanoparticles exhibited synergistic cytotoxicity with improved anti-tumor activity in the orthotopic TNBC mouse model	[191]
11	TPGS coated liposomes	EGFR inhibitor: Cetuximab	Paclitaxel and piperine	- The targeted liposomes showed increased cellular uptake and cytotoxicity compared to non-targeted liposomes and free drugs	[192]
12	Albumin nanoparticles	Folic acid	Artemether	- FRα overexpressed TNBC cells showed increased cellular internalization of folate-targeted nanoparticles - The nanoparticles also showed increased cytotoxicity and apoptosis	[179]
13	Polymeric nanoparticles	hyaluronic acid	Thymoquinone	- The targeted nanoparticles diminished the cell migration by upregulating microRNA-361, which in turn downregulated Rac1 and RhoA - The nanoparticles also decreased the secretion of VEGFR-A, which in turn decreased angiogenesis and metastasis	[180]
14	Polymeric micelles	Inhibitor of tyrosine kinase receptor: sunitinib	Paclitaxel	- The nanoparticles significantly improved the ICD response, which in turn resulted in increased tumor immunogenicity and apoptosis	[181]
15	Macrophage-derived exosomes	c-Met binding peptide	Doxorubicin	- The exosomes increased the cellular internalization and anti-tumor activity - The nanoparticles also increased the targetability, in addition to increased apoptosis and decreased tumor growth	[193]
16	Chitosan nanoparticles	PARP inhibitor: olaparib, and FOXM1-siRNA	Paclitaxel	- The nanoparticles showed increased targetability and cytotoxicity	[182]
17	Cancer cell membrane (CM)-cloaked upconversion nanoparticles,	anti-CD73 antibody	ROS-sensitive polymer polyethylene glycol-thioketal-doxorubicin	- The presence of anti-CD72 antibody in the nanoparticles inhibited the immunosuppressive activity by disrupting the adenosine pathway - The nanoparticles also showed increased synergistic anticancer activity - Additionally, the nanoparticles inhibited the abscopal tumor metastasis by blocking CD73 responses	[198]
18	Chitosan—lactate nanoparticles	HIV-1 derived TAT peptide and CD73 siRNA	Doxorubicin	- The nanoparticles caused apoptosis and inhibited the proliferation and migration of cancer cells - Moreover, the preferential internalization of NPs decreased cancer growth, proliferation, and migration, and increased the survival rate of the tumor-induced mice	[183]
19	Polymeric nanoparticles	PD-L1 blocking antibodies	CD155 siRNA	- The nanoparticles increased the CD8+ T cell immune surveillance in early-stage TNBC - Moreover, the co-delivery enhanced the immune checkpoint therapy - Nanoparticles also showed inhibition of the progression and metastasis of TNBC cells	[184]
20	Ferritin nanoparticles	EGFR inhibitor: lapatinib	Pseudolaric acid B	- The co-loaded ferritin nanoparticles showed decreased cytotoxicity with inhibition of proliferation and migration	[171]
21	Solid lipid nanoparticles (SLNs)	anti-EGFR/CD44 dual-RNA aptamers,	Doxorubicin	- The targeted nanoparticles inhibited cancer cell proliferation	[194]
22	Polymeric micelles	PARP inhibitor; olaparib (OLA)	Dasatinib	- The nanoparticles showed increased cellular uptake by surface binding with the overexpressed CD44 proteins of TNBC cells - The nanoparticles also showed prolonged circulation time, thereby increasing the bioavailability and improving the anticancer activity	[185]
23	Lipid nanoparticle	ICAM—1 binding peptide, LFA1–P	Gemcitabine and Paclitaxel	- The nanoparticle showed increased cellular accumulation - The targeted nanoparticles showed a 60-fold-increased target/healthy tissue (lung/GI) ratio compared to non-targeted nanoparticles	[195]

**Table 4 cancers-15-02661-t004:** Clinical trials of biomarker-based nanoparticles against TNBC.

Formulation	Drug	Target/Biomarker	Ligand	Clinical Phase (NCT Number)
Glembatumumab—Vedotin-antibody drug conjugate	MMAE (auristatin)	NMB glycoprotein	Glembatumumab (anti NMB glycoprotein monoclonal antibody)	Phase II (NCT01997333)
Cofetuzumab—pelidotin (PF-06647020) Albumin nanoparticles	Aur001 (auristatin)	PTK7 (protein tyrosine kinase—7)	Cofetuzumab (anti-PTK7 monoclonal antibody)	Phase I (NCT03243331/NCT02222922)
PF-06647263—Albumin nanoparticles	Calicheamicin	Ephrin receptor-4	Anti-Ephrin receptor-4 monoclonal antibody	Phase I (NCT02078752)
Nab-rapamycin—Albumin nanoparticles	Rapamycin	gP 60 receptors	Albumin	Phase I (NCT02646319)
C225-ILS-Dox—liposomes	Doxorubicin	EGFR	Antigen-binding fragment of cetuximab	Phase II (NCT02833766)
MM310—liposomes	Docetaxel pro-drug	Ephrin A2	Anti-ephrin A2 monoclonal antibody	Phase I (NCT03076372)

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
