# Peer review of "Endorsement of TNBC Biomarkers in Precision Therapy by Nanotechnology"

_cancers, 2023, doi:10.3390/cancers15092661_

Round 1

Reviewer 1 Report

The present manuscript entitled “Endorsement of TNBC biomarkers in precision therapy by the nanotechnology” investigated the application of various biomarkers in the field of

targeted therapy and their delivery through nanoparticles.

There are some specific comments that need to addressed as follows:

Comment 1: The author has mentioned various biomarkers that are co-related with TNBC progression, however, the author had missed some common biomarkers that also plays a rolein targeting, namely claudin protein, and claveolin. It is advised to add a sub-section discussing the role of such proteins as biomarkers for TNBC.

Comment 2: The authors should add a table indicating the clinical studies of various biomarkers of TNBC, as this will provide more knowledge regarding the current scenario of the biomarker’s application clinically.

Comment 3: From various studies, it came into knowledge that nanoparticles form a“protein-corona” when comes in contact with biological fluids, that affects the nanoparticle-cell interaction and influence cellular uptake, accumulation, degradation, and clearance. The authors thus should incorporate a paragraph discussing the subject stated above for better understanding the fate of the nanoparticles inside the biological fluids.

Author Response

The present manuscript entitled “Endorsement of TNBC biomarkers in precision therapy by the nanotechnology” investigated the application of various biomarkers in the field of targeted therapy and their delivery through nanoparticles.

There are some specific comments that need to addressed as follows:

Comment 1: The author has mentioned various biomarkers that are co-related with TNBC progression, however, the author had missed some common biomarkers that also plays a role in targeting, namely claudin protein, and caveolin. It is advised to add a sub-section discussing the role of such proteins as biomarkers for TNBC.

Response 1: We thank the reviewer for drawing our attention. As suggested, we have added the sub-sections indicating the role of claudin and caveolin proteins as a biomarker for TNBC in the revised manuscript.

         Claudins are tight junctional proteins existing between the epithelial cells and creating a barrier for the transport of macromolecules. However, in neoplastic cells, these tight junctions experience structural and functional defects which destroy them [63]. From various studies, it was observed that 66.1% of TNBC cases show increased expression of claudin-4, along with an evident positive correlation with tumor size, nodal status, metastasis, and an expression of Ki-67 [63]. Claudin proteins are also considered prognostic-type biomarkers. In recent times, it was found that in addition to claudin-4, claudin-3, and claudin-7 are also regarded as promising prognostic factors in TNBC, which was found relevant through their aberrant immunohistochemical expressions. It was further documented that increased expression of claudin-3 was correlated with the mutation of BRCA1 genes which further aid in testing the BRCA mutation for TNBC patients [64]. There is another claudin protein named claudin -1 which unlike above mentioned claudins, serves as tumor suppressor in TNBC. It was documented that re-surfacing of claudin-1 on TNBC cells induces apoptosis. From various clinical studies, it was observed that loss of expression of claudin-1 is associated with malignancy, invasiveness, and recurrence of TNBC [65]

Caveolins (Caveolin -1, 2, and 3) are scaffold proteins composed of cholesterol-enriched microdomains and play an important role in tumor progression. It was further found that among various caveolins, caveolin-1 (Cav 1) plays a potential role in membrane trafficking, cell invasion, and proliferation, cell migration, cell metastasis, and apoptosis, and belongs to the prognostic type of biomarker. However, it was found that depending on the subtypes of cancers, the caveolin-1 can function either as a tumor suppressor or a tumor promoter. Cav-1 functions as an anti-proliferative factor in TNBC by arresting the cell cycle at the G2/M phase, which can be promoted by upregulating certain tumor suppressor genes namely p21, p27, and downregulating cyclin D2 [66].  In recent data, it was observed that the loss of normal Cav-1 is linked with phosphorylation of AKT, TGF-β1, and acceleration of the aggressiveness of TNBC [67].

Comment 2: The authors should add a table indicating the clinical studies of various biomarkers of TNBC, as this will provide more knowledge regarding the current scenario of the biomarker’s application clinically.

Response 2: As per the suggestion, we have added a table indicating the clinical trials performed on various biomarkers for the diagnosis and treatment of TNBC.

In table 1, we have listed various clinical studies that are either completed or ongoing involving the participation of biomarkers in the treatment of TNBC.

Comment 3: From various studies, it came into knowledge that nanoparticles form a “protein-corona” when comes in contact with biological fluids, that affects the nanoparticle-cell interaction and influence cellular uptake, accumulation, degradation, and clearance. The authors thus should incorporate a paragraph discussing the subject stated above for better understanding the fate of the nanoparticles inside the biological fluids.

Response 3: The suggestion was well taken. In the revised manuscript, we have added a section demonstrating the impact of “protein corona” on the fate of nanoparticles and the ways to overcome its influence over the nanoparticles.

Although the NPs emerged as an efficient drug delivery system over conventional delivery systems due to their above-mentioned characteristics and advantages, a certain fraction of them reaches the clinics. It was further found that due to the presence of a mononuclear phagocytic system, most of the NPs get accumulated in off-target organs after administration, as compared to the target site. One of the reasons for such a scenario is the prevalence of “protein corona (PC)” which is described as the array of proteins attached to NPs, affecting its colloidal stability, optimal bio-distribution, interactions, and clearance [184]. Moreover, it was established that the assembly of PC and NPs was the first interaction that the NPs experience after entering the body. So, to overcome such interactions, various studies were performed, where PEGylation was found to be the widely used strategy. But it also showers some hindrance which includes a decrease of cellular uptake of NPs by the target cells, and the formation of circulating anti-PEG antibodies on excessive administration of PEGs, thus creating a state of hypersensitivity, and compromising the safety of NPs [185]. Henceforth, such synthetic strategies were overtaken by the biomimetic approaches which include a coating of NPs with the cell membrane, application of the virus or its components, and manipulating the integrity of PC [186].  

It was further revealed that coating the NPs with cell membrane makes the NPs, an inherent biological identity, which aids the NPs in avoiding opsonization as well as production of anti-NP immunoglobulins that further facilitates rapid clearance of NPs [181]. It was not far unnoticed that viruses can be considered in the design of drug delivery systems. Viruses and their components were found to interact with the proteins in the biological fluids. Berardi, et al., 2019 found that cowpea mosaic virus (CPMV) coated NPs and bluetongue virus (BTV) coated NPs exhibited the existence of less PC, compared to standard polystyrene NPs [184].  Also, it was found that apart from a coating of NPs with the cell membrane, and viruses, the manipulation of PC attached to NPs also plays an important role in diminishing its interaction with NPs. The PC environment can be manipulated by attaching the NPs with various stealth-inducing biological materials like alginic acid or hyaluronic acid. Such attachment alters the size, surface potential, and stability of the NPs [187].

Hence, it could be inferred that the interaction of PC and NPs can be manipulated by understanding the composition, and integrity of the PC developed around the NPs. In this perspective, it becomes essential to develop a specific harmonized technique for analyzing the assembly of PC.

Reviewer 2 Report

It’s nothing wrong with the framework of this review for clearly illustrating the contents: the Biomark-based Targeted therapy, TNBC biomark's role in cancer nanotherapeutics and clinical citation. But this review paper is not professional or critical in reviewing the published literature. Authors are only describing and summarizing the reported results.  

1.The introduction part was written too simple. It just like simply list all related factor and lacking of critical thinking.

2. Authors are only describing and summarizing the reported results in the second part. It’s a comprehensive review listing 30 biomarkers for TNBC molecular analysis. Can it be classified? Can the authors select typical or the important factor to further discussion? It should be more useful.

3.The fourth part and the fifth part are still widely introduced. The content is broad and unfocused. The reviewer/reader cannot figure it out any useful strategies for TNBC clinical treatment. It is suggested to sort out a main line and give some available information. These two parts seem to be a list of a lot of literature reading, without authors’ own critical thinking.

 4.Conclusion and Future perspective are too simple. Authors’ own idea is needed.

 5.Other details need to be paid attention to while rewriting.

(1) Some pictures look a little fuzzy and are difficult to recognize, please provide the sharp and clear version. Eg. Fig3 part 1 and part 2c, Fig7c,

(2) The Fig legend need to be double check. For example, the legend in Figure 5 has errors.

(3) Please notice how the paragraphs are formatted. Some paragraphs don't start with a space.

Author Response

It’s nothing wrong with the framework of this review for clearly illustrating the contents: the Biomarker-based Targeted therapy, TNBC biomarker's role in cancer nanotherapeutics and clinical citation. But this review paper is not professional or critical in reviewing the published literature. Authors are only describing and summarizing the reported results.  

Comment 1: The introduction part was written too simple. It just like simply list all related factor and lacking of critical thinking.

Response 1: The suggestions are well taken. As per the suggestion, we have modified the introduction part of the revised manuscript.

Triple-negative breast cancer (TNBC) is the most aggressive subtype of breast cancer with no expression of estrogen receptor, progesterone receptor, and human epidermal receptor -2. It accounts for approximately 10 – 20% of total breast cancer cases and is found to be more prevalent in African and Hispanic young women [1]. According to the American Cancer Society, and National Cancer Institute, in the year 2020, approximately 276,480 new cases of TNBC occurred, where almost 42,170 women got deceased [2]. TNBC is considered aggressive due to its heterogeneity, rapid metastasizing ability to the brain, lungs, and bones, and rapid onset of recurrence [3]. The such scenario makes the treatment regimen difficult for TNBC. Also, as TNBC lacks the expression of hormones, endocrine therapy is out of the option, which makes chemotherapy the only treatment against TNBC [4]. From the molecular profiling, it came into focus, that there are six molecular subtypes of TNBC, which are basal-like subtypes (BL1, and BL2), mesenchymal (M), mesenchymal stem-like (MSL), immunomodulatory (IM), and luminal androgen receptor (LAR) [5]. Further, on performing the genetic profiling of the molecular subtypes, it was found that these subtypes show either aberrant genetic expression or highly activated signaling pathways or receptors. For example, BL1, and BL2 subtype shows an aberrant expression of DNA-repair and cell-cycle regulating-related genes like MYC, PIK3CA, AKT2, CDK6, BRCA2, PTEN, RB1, and TP53. Similarly, MSL subtypes also show aberrant expression of genes related to cell proliferation, and stemness (ALDHA1, BCL2, BMP2, HOX, etc.). On the other hand, M and IM subtypes exhibit highly activated signaling pathways like Wnt, TGF-β, NK cell, IL-12, IL-7, etc. Moreover, LAR subtypes show highly activated androgen hormone-related signaling pathways [6]. It was thus inferred that the heterogeneous nature of TNBC might compromise the therapeutic efficacy of the chemotherapy. Moreover, the conventional neoadjuvant chemotherapy exhibited pCR in 35 – 45% of TNBC patients only, and the majority of the TNBC patients showing responsiveness to conventional chemotherapy were limited to the non-metastatic stage [7]. Thus, to overcome such problems and make the treatment more precise, biomarkers have emerged as the targeted therapeutic and diagnostic tool. In recent times, scientists are using cancer biomarkers to acquire knowledge regarding the patient’s tumor to predict the personalized treatment regimen, specific to a particular TNBC subtype. These predictive biomarkers include various germline and somatic mutations, genetic rearrangements, proteins, and metabolomics [8]. However, it was observed that none of the biomarkers achieve 100% sensitivity as well as specificity [9], and also, as the cancer treatment implements more of combination therapy, compared to monotherapy, it becomes difficult to attach the identified biomarker with the single drug or target [10]. Hence, to increase the specificity, and to efficiently deliver multiple diagnostic and therapeutic molecules to the target site, nanoparticles (NPs) were developed based on their exclusive physiochemical characteristics. It was further observed that for improved sensitivity, and targetability, the NPs are modulated to incorporate cancer-specific ligands having an increased binding affinity towards TNBC biomarkers [9].

In this review, we discuss well-established TNBC biomarkers and explored nanoparticle-based technologies employed for increasing the sensitivity, and specificity of the biomarkers toward the targeted site. We also discuss the ongoing clinical trials on the biomarkers, and biomarkers-based nanoparticles, employed as a therapeutic and diagnostic tool against TNBC.

Comment 2. Authors are only describing and summarizing the reported results in the second part. It’s a comprehensive review listing 30 biomarkers for TNBC molecular analysis. Can it be classified? Can the authors select typical or the important factor to further discussion? It should be more useful.

Response 2: As suggested by the reviewer, we have classified the biomarkers and added a section indicating the important factors to be considered while targeting biomarkers.

In recent times, a series of TNBC-biomarkers have been evaluated. The TNBC biomarkers can be classified based on usages like prognostic (biomarkers giving information regarding the overall outcome, regardless of the therapy), predictive (biomarkers giving information on the effect of the therapeutic intervention), and diagnostic (biomarkers confirming the presence of the disease) [13], or on the site, where the biomarkers are found of availability like in the blood, cytoplasm, nucleus, and on the surface of cells [10], or on target expression like DNA, RNA, and proteins [14]. It was found that one biomarker can be prognostic, predictive, and diagnostic at the same time [15]. It becomes reasonable to classify biomarkers based on the site, where they are found, as it will aid in developing a strategy suitable for delivering the diagnostic and therapeutic moiety to the target site more efficiently. Moreover, if their site of availability is known, we can modulate or personalize the delivery system or targeting system by changing their nature, and characteristics.

Although we have described the role of each biomarker in the progression of TNBC, it was recently observed that scientists are endorsing a combination of biomarkers for better and more efficient results. In such a scenario, the specificity, as well as sensitivity of individual biomarkers, are kept optimum, so that, it could result in a better prognosis or effective diagnosis. Hence, before selecting a biomarker or combination of biomarkers, the family history of the patient, and the lifestyle should be taken into consideration. Also, certain ideal characteristics of biomarkers recorded include the expression of biomarkers in the early stage of the disease, able to discriminate diseased populations from the healthy population, and can reproduce results. All these factors and criteria result in effective therapy and diagnosis [108].      

Comment 3. The fourth part and the fifth part are still widely introduced. The content is broad and unfocused. The reviewer/reader cannot figure it out any useful strategies for TNBC clinical treatment. It is suggested to sort out a main line and give some available information. These two parts seem to be a list of a lot of literature reading, without authors’ own critical thinking.

Response 3: As suggested by the reviewer, we have modified the fourth part of the revised manuscript and added a new paragraph indicating the latest advances in nanotherapeutics.  

In recent times, scientists are focusing more on green chemistry and green science, where more sustainable and natural compounds are synthesized, thereby mimicking biological or living conditions to a maximum extent. In recent times, natural materials are combined with synthetic NPs to bio-mimic the nanoparticles into the biological system, leading to increased drug-targetability, and reduced immunogenicity. In this context, researchers are extracting natural cell membranes and coating synthetic NPs with them, thus, facilitating enhanced biofunctionalization. It was observed that cell membrane coating preserved the physiochemical integrity of the NPs while exerting intrinsic and complex cellular functions. Also, it was observed that for mediating enhanced targeting, the NPs are coated with the membrane derived from that specific cell, and such targeting was observed through homotypic, and heterotypic adhesion [181]. Further, it was summarized that the cell membrane-coated NPs facilitate advanced immune responses, bypass clearance via RES, prolonged circulation, and enhanced targetability [182].

Although the NPs emerged as an efficient drug delivery system over conventional delivery systems due to their above-mentioned characteristics and advantages, a certain fraction of them reaches the clinics. It was further found that due to the presence of a mononuclear phagocytic system, most of the NPs get accumulated in off-target organs after administration, as compared to the target site. One of the reasons for such a scenario is the prevalence of “protein corona (PC)” which is described as the array of proteins attached to NPs, affecting its colloidal stability, optimal bio-distribution, interactions, and clearance [184]. Moreover, it was established that the assembly of PC and NPs was the first interaction that the NPs experience after entering the body. So, to overcome such interactions, various studies were performed, where PEGylation was found to be the widely used strategy. But it also showers some hindrance which includes a decrease of cellular uptake of NPs by the target cells, and the formation of circulating anti-PEG antibodies on excessive administration of PEGs, thus creating a state of hypersensitivity, and compromising the safety of NPs [185]. Henceforth, such synthetic strategies were overtaken by the biomimetic approaches which include a coating of NPs with the cell membranes, application of viruses or their components, and manipulating the integrity of PC [186].  

It was further revealed that coating the NPs with cell membrane makes the NPs, an inherent biological identity, which aids the NPs in avoiding opsonization as well as production of anti-NP immunoglobulins that further facilitates rapid clearance of NPs [181]. It was not far unnoticed that viruses can be considered in the design of drug delivery systems. Viruses and their components were found to interact with the proteins in the biological fluids. Berardi, et al., 2019 found that cowpea mosaic virus (CPMV) coated NPs and bluetongue virus (BTV) coated NPs exhibited the existence of less PC, compared to standard polystyrene NPs [184].  Also, it was found that apart from the coating of NPs with the cell membrane, and viruses, the manipulation of PC attached to NPs also plays an important role in diminishing its interaction with NPs. The PC environment can be manipulated by attaching the NPs with various stealth-inducing biological materials like alginic acid or hyaluronic acid. Such attachment alters the size, surface potential, and stability of the NPs [187].

Hence, it could be inferred from the above paragraph that the interaction of PC and NPs can be manipulated by understanding the composition, and integrity of the PC developed around the NPs. In this perspective, it becomes essential to develop a specific harmonized technique for analyzing the assembly of PC.

Comment 4. Conclusion and Future perspective are too simple. Authors’ own idea is needed.

Response 4: As suggested by the reviewer, we have modified the conclusion and future perspective in the revised manuscript.

From the present review, we concluded that an extensive advancement has been made to understand the biology of cancer which further led to the recognition of the heterogeneous nature of TNBC. Such heterogeneous characteristics aroused due to the presence of six subtypes of TNBC, which further led to the identification of various prognostic, and predictive biomarkers, as evidenced by the activation or inactivation of distinct signaling pathways, receptors, genes, etc.  As each subtype is characterized by specific biomarkers, hence targeting those biomarkers resulted in precise and personalized TNBC therapy, thereby avoiding intraspecies variability. Further, biomarker screening has been proven an important strategy for the characterization of distinct cancer types and cellular aberrations that trigger the development of cancer. However, despite such advancements, it was demonstrated that chemotherapy remained the backbone of TNBC therapy, as they are insensitive to endocrine treatment. So, to broaden the treatment regimen of TNBC, and escape the off-target side effects of chemotherapy, various targeted therapies were adopted based on the biomarkers. In the present article, we have discussed various signaling pathways, aberrant gene expressions, and receptors, responsible for TNBC progression, as well as the therapeutics assigned for their treatment.

It was further observed that the conventional drug delivery system proved inefficient in delivering targeted therapy, moreover, it has its drawbacks like lower efficacy, non-tunable, etc. All these characteristics are observed in the case of nanoparticles which offer surface modification or functionalization, increased targeting, and enhanced efficacy. The application of nanotechnology in the precise delivery of anticancer drugs, molecules like DNA, siRNA, etc further enhances the chances of therapeutic success, as well as helps in integrating the molecular biomarkers for distinctive recognition of biomolecules that are specific to each subtype of TNBC. It was thus concluded that conjugating these biomarkers to nanotechnology-based platforms critically improves the efficiency of anticancer therapy and improves TNBC management.

Despite the advantages of nanoparticles in the treatment and diagnosis of TNBC, very limited nanotherapeutics find their way to the clinics, mostly due to their complex design, lack of expertise in the manufacturing sector, and regulating guidelines, cost, and testing parameters. It was further suggested that as nanotherapeutics are emerging as a pivotal platform for diagnosis, and treatment, updated evaluation and policy regarding nanotherapeutics are being established, clearing the roadblock responsible for not letting the nanotherapeutics into the clinics. Also, it was observed, as artificial intelligence (AI) is getting associated with every field of life, it was assumed that soon, scientists will be incorporating artificial intelligence technologies in nanotherapeutics as well as biomarkers screening and targeting. AI will be employing extensive algorithms in extracting more precise and exhaustive information regarding the biomarkers present in the heterogenous population of TNBC patients. It will save time, manpower, and source in doing the same. We thus believe that, in the coming future, AI will help in identifying more new biomarkers and developing the latest biomarkers-driven nanotherapeutics. Such innovations led to the development of bioinformatics. It was observed that with bioinformatics, one can predict the origin of cancer, and its mRNA expression, thus paving the way to a whole new world of cancer-genomic, which will later aid in developing new drugs and their delivery system

 Comment 5. Other details need to be paid attention to while rewriting.

  • Some pictures look a little fuzzy and are difficult to recognize, please provide the sharp and clear version. Eg. Fig3 part 1 and part 2c, Fig7c,

Reponses: The images were taken from published reports and hence we have limitations in further improving the quality.

  • The Fig legend need to be double check. For example, the legend in Figure 5 has errors.

Responses: We have double-checked the legends and corrected wherever necessary.

  • Please notice how the paragraphs are formatted. Some paragraphs don't start with a space.

Responses: We have followed the journal guidelines regarding the same.

Reviewer 3 Report

The most frequent cancer in women is breast cancer. TNBC is thought to be a rare and aggressive subtype of breast cancer. The only treatment for TNBC is still chemotherapy. The emergence of inherent or acquired chemo-resistance limited the treatment of TNBC. The authors of this article address the different possible biomarkers for TNBC treatment, as well as their application in nanotechnology in TNBC management and therapy. However, I have major concerns about this article.

The way the authors described cellular signals in this review is widely recognized. The authors should outline the relevant gaps and issues and offer fresh viewpoints. Additionally, they should create clear research directions. The review should highlight the heavily debated and current subject; this article appears as a book chapter more than a review.

Despite the author's inclusion of the copyright figures. Even though the authors exact copied the figure legends from the articles. Using the authors' own article figures and legends is advised. 

Figures 1 and 2 should be described in the legends.

The described cellular signal in the figures should be modified because it is generally known. In addition, they should have mentioned the figure numbers in the text. There are no matches of the figures within described text. 

In some places, the authors only paraphrased from a single article. For example 

1. Introduction refrence1

2.3 TP53 reference 6,7

2.18. ETS translocation variant4 (ETV4) reference 61

2.20. G–protein-coupled receptor 161 (GPR161) reference 65

2.21. G–protein-coupled Kisspeptin receptor (KISS1R) reference 66

2.22. TNBC biomarkers in cancer nanotherapeutics reference 124.

Above are just examples; this was found throughout the text in many paragraphs. 

Author Response

The most frequent cancer in women is breast cancer. TNBC is thought to be a rare and aggressive subtype of breast cancer. The only treatment for TNBC is still chemotherapy. The emergence of inherent or acquired chemo-resistance limited the treatment of TNBC. The authors of this article address the different possible biomarkers for TNBC treatment, as well as their application in nanotechnology in TNBC management and therapy. However, I have major concerns about this article.

Comment 1: The way the authors described cellular signals in this review is widely recognized. The authors should outline the relevant gaps and issues and offer fresh viewpoints. Additionally, they should create clear research directions. The review should highlight the heavily debated and current subject; this article appears as a book chapter more than a review.

Response 1: The suggestions are well taken. We have tried our best to modify the manuscript as per the suggestions.

Comment 2: Despite the author's inclusion of the copyright figures. Even though the authors exact copied the figure legends from the articles. Using the authors' own article figures and legends is advised.

Response 2: Our findings are under consideration in some other journals and have not been published yet. 

Comment 3: Figures 1 and 2 should be described in the legends.

Response 3: As suggested by the reviewer, we have described Figures 1, and 2 in the legends.

Comment 4: The described cellular signal in the figures should be modified because it is generally known. In addition, they should have mentioned the figure numbers in the text. There are no matches of the figures within described text. 

Response 4: In the figure, we are just explaining the general pathways responsible for the progression of TNBC.

We have mentioned the figure number in the text of the revised manuscript.

Comment 5: In some places, the authors only paraphrased from a single article. For example 

  1. Introduction refrence1

2.3 TP53 reference 6,7

2.18. ETS translocation variant4 (ETV4) reference 61

2.20. G–protein-coupled receptor 161 (GPR161) reference 65

2.21. G–protein-coupled Kisspeptin receptor (KISS1R) reference 66

2.22. TNBC biomarkers in cancer nanotherapeutics reference 124.

Above are just examples; this was found throughout the text in many paragraphs.

Response: We have modified the paragraphs in the revised manuscript.

Reviewer 4 Report

Thank you for submitting this comprehensive manuscript however it was too long for reader to follow.  I would suggest to shorten it to 25-30 pages with more focus on nanotechnology.  Most of the information about biomarkers are well known and can be removed with more focus on clinical implications rather than biology.

Author Response

Comment 1: Thank you for submitting this comprehensive manuscript however it was too long for reader to follow.  I would suggest to shorten it to 25-30 pages with more focus on nanotechnology.  Most of the information about biomarkers are well known and can be removed with more focus on clinical implications rather than biology.

Response: The suggestion is well-taken. We tried to shorten the pages with more focus on nanotechnology, and its clinical implications, in comparison to general biology.

Round 2

Reviewer 2 Report

The authors have answered all my questions. After some editing of English language, the review can be accepted.

Author Response

Comment 1: The authors have answered all my questions. After some editing of English language, the review can be accepted.

Response: We thank the reviewer for the time and effort spared to review the manuscript and the constructive suggestions. We have gone through the manuscript once again to uplift the English as much as possible.

Reviewer 3 Report

The authors have improved the articles. However, this article still contains several significant flaws, though. I suggested that the authors draw attention to the recent controversial topic of breast cancer nanotherapy. Except for a few references, the updated article looked exactly the same as before.

The issues covered in the articles are not novel to the readers and fail to demonstrate any advancements in the field. Numerous review articles have reported on various drug delivery methods for treating breast cancer. The author should discuss the most recent research on nanoparticles and breast cancer, the constraints of drug delivery failure, and strategies for overcoming these gaps—for example, PMID: 35236941, and how the nano therapy targets the tumor microenvironment.

Furthermore, there is still a problem with the phrases in the updated article. An example. References 95 and 97 deal with activating transcription factor 4 (ATF4), and ETVA, respectively.

Because this is not a book chapter, I suggest the authors delete the copyright figures. This dilutes the review article's content.

Author Response

Comment 1: The author should discuss the most recent research on nanoparticles and breast cancer, the constraints of drug delivery failure, and strategies for overcoming these gaps

Response: As suggested by the reviewer, we have added a section in the revised manuscript indicating the constraints of drug delivery failure, and strategies for overcoming these gaps.  

Drug delivery refers to delivering pharmaceuticals, small molecules, genes, and biomolecules to the diseased site (cell or organ), facilitating desired therapeutic effects and minimizing side effects [148]. However, it was observed that 80% of the clinical drugs fail to produce desired therapeutic efficacy [149], as they suffer from insufficient bioavailability due to poor water solubility, permeability, and biological barriers. Thus, it was demonstrated that therapeutic performance does not depend merely on the activity of the administered moieties but also their bioavailability at the target site. Furthermore, it has been observed that conventional drug delivery systems demonstrate severe constraints like non-controlled drug release, non-targeting delivery, systemic side effects, increased and frequent dosing, and poor bioavailability [150].

         Various encouraging drug delivery strategies have been evolved to overcome the failure of conventional drug delivery systems, such as drug optimization, drug modifications, microenvironment modification, and the emergence of novel drug delivery systems. In drug optimization, structure‒tissue exposure/selectivity activity relationship (STAR) was employed, which improves the drug optimization, by classifying the drug candidates based on the potency, tissue exposure, selectivity, required dose for balancing clinical efficacy, and toxicity. Such an approach overcomes the gaps caused by the structure-activity relationship (SAR), which only indicates potency/specificity. It overlooks tissue exposure/selectivity in disease/normal tissues, thereby misleading the selection criteria of the drug candidate selection and their impact on the balance between efficacy and toxicity [149]. In drug modification, the structure of the drug gets altered by changing the orientation, nature, or type of functional groups, amino acids, or nucleic acid backbones, thereby improving the pharmacokinetic attributes. Also, the drug can be conjugated with known moieties and targeting ligands, thereby increasing the targetability and therapeutic efficacy and improving the drug release profile. Such a strategy aims to modulate the interaction of the drug and the tissues or cells and control the navigation of the drug from its administration to the desired therapeutic activity. In microenvironment modification, the host environment gets altered, which significantly changes the mechanistic approach of the drug at the site of action, like applying pH modifiers, permeation enhancers, protease inhibitors, enzyme inhibitors, etc. Such microenvironment modification aid in navigating biological barriers [151]. Lastly, pharmaceutical companies are investing more in developing novel drug delivery systems that exhibit excellent therapeutic performance, flexible drug release profile as per the desired disease, clinical efficacy, prolonged product life, increased targetability, reduced dose frequency, and dose-dependent side effects [150].

            Nanoparticles have shown evidence that showcases their efficacy in bypassing the limitations of the conventional drug delivery system, such as biodistribution and intracellular trafficking via site-specific targeting. With this realization, the US National Science and Technology Council (NSTC) launched an initiative named National Nanotechnology Initiative (NNI) in 2000 that outlined the efforts to improve therapeutic research through the emergence of nanotechnology [152].

Comment 2: How the nanotherapy target the tumor microenvironment?

Response: As suggested by the reviewer, we have added a section in the revised manuscript indicating the existence of a tumor microenvironment (TME), and discussed the targeting phenomenon of nanoparticles to the tumor microenvironment. 

Tumor microenvironment (TME) is also gaining much attention in treating cancer cells by nanoparticles. It has been observed that the TME comprises cellular and structural components like fibroblast, extracellular matrix, immune cells, and vasculature that surround the cancer sites and help in the growth of cancer cells and their metastasis. It was further found that the existence of TME limits the delivery of chemotherapy to the cancer site, thereby leading to the failure of the therapy. However, recent treatments like antiangiogenic therapy and immunostimulatory therapy showed limited success despite demonstrating encouraging pre-clinical results. Such limitations are due to the lack of drug penetration into the necrotic tumor core, non-specific delivery, rapid elimination from serum, and dose-depended toxicity. All these problems were further resolved by applying nanoparticles that target the TME vasculature, ECM, and immune response [159]. Usually, while targeting the TME, various pathophysiological conditions of TME are taken care of, including enzymatic activity, hypoxia, oxidative stress, high interstitial fluid pressure, levels of amino acids, functional proteins, levels of macrophages, lymphocytes, and neutrophils. Pre-clinically, it was observed that the NPs target TME by involving pegylated, stimuli-responsive, and dual-functional nanoparticles. More specific strategies involve site-specific attachment of PEG linkage, surface-charge reversal, decrease in particle size, hyperthermia-induced generation of CO2, response to internal stimuli like pH, temperature, and external stimuli like a magnetic field, light, ultrasound, etc. [160]. Various studies showed that encapsulating cytokines, siRNA, and cytotoxic drugs within nanoparticles induces immune stimulation, which can either kill or modify the tumor-associated macrophages, an important component of TME. Additionally, nanoparticles can target distinct immune subpopulation like T cells, NK cells, and DCs through surface functionalization [159]. Besides this, the upsurging understanding and knowledge of targeting TME using nanoparticles are paving the way for the fabrication of a combined strategy involving therapeutics and diagnostics, commonly known as nanotheranostics

Comment 3: Furthermore, there is still a problem with the phrases in the updated article. An example. References 95 and 97 deal with activating transcription factor 4 (ATF4), and ETVA, respectively.

Response: We tried to solve the phrase issue as much as possible. However, it is suggested to look upon the reference number as the activating transcription factor 4 (ATF4) deals with ref# 97, and 98, and ETVA deals with ref # 99, and 100.

Ref # 95 discusses the impact of TP53.

Comment 4: Because this is not a book chapter, I suggest the authors delete the copyright figures. This dilutes the review article's content.

Response: We are sorry but not convinced by the reviewer’s suggestion. As a reader, we get a more elaborate understanding of the topic which is being discussed in any review article. As per our understanding this will provide a practical view point to the readers rather than diluting the content. However, we don’t have any reservations and are ready to remove all the figures (or the specifically suggested figures) upon further suggestion by the Editor. 
